# IGF-1 facilitates extinction of conditioned fear

**Laura E Maglio**[1,2†]\*, **José A Noriega-Prieto**[1,3†], **Irene B Maroto**[1,4], **Jesús Martin-Cortecero**[1,5], **Antonio Muñoz-Callejas**[1], **Marta Callejo-Móstoles**[1], **David Fernández de Sevilla**[1]\*

[1]Departamento de Anatomía, Histología y Neurociencia, Facultad de Medicina, Universidad Autónoma de Madrid, Madrid, Spain; [2]Departamento de Ciencias Médicas Básicas (Fisiología) and Instituto de Tecnologías Biomédicas (ITB), Universidad de La Laguna, Tenerife, Spain; [3]Department of Neuroscience, University of Minnesota, Minneapolis, United States; [4]Centro de Investigación Biomédica en Red Sobre Enfermedades Neurodegenerativas (CIBERNED), Instituto Universitario de Investigación Neuroquímica (IUIN), Instituto Ramón y Cajal de Investigación Sanitaria (IRYCIS) and Departamento de Bioquímica y Biología Molecular, Facultad de Química, Universidad Complutense de Madrid, Madrid, Spain; [5]Institute of Physiology and Pathophysiology, Medical Biophysic, Heidelberg University, Heidelberg, Germany

**Abstract** Insulin-like growth factor-1 (IGF-1) plays a key role in synaptic plasticity, spatial learning, and anxiety-like behavioral processes. While IGF-1 regulates neuronal firing and synaptic transmission in many areas of the central nervous system, its signaling and consequences on excitability, synaptic plasticity, and animal behavior dependent on the prefrontal cortex remain unexplored. Here, we show that IGF-1 induces a long-lasting depression of the medium and slow post-spike afterhyperpolarization (mAHP and sAHP), increasing the excitability of layer 5 pyramidal neurons of the rat infralimbic cortex. Besides, IGF-1 mediates a presynaptic long-term depression of both inhibitory and excitatory synaptic transmission in these neurons. The net effect of this IGF-1-mediated synaptic plasticity is a long-term potentiation of the postsynaptic potentials. Moreover, we demonstrate that IGF-1 favors the fear extinction memory. These results show novel functional consequences of IGF-1 signaling, revealing IGF-1 as a key element in the control of the fear extinction memory.

\*For correspondence:
lamaglio@ull.edu.es (LEM);
david.fernandezdesevilla@uam.es
(DF)

†These authors contributed equally to this work

**Competing interests:** The authors declare that no competing interests exist.

## Introduction

Conditioned fear is an associative form of learning and memory in which a previous neutral stimulus (called 'conditioned stimulus' [CS]; e.g., a tone) comes to elicit a fear response when is associated with an aversive stimulus (called 'unconditioned stimulus' [US]; e.g., an electric shock *LeDoux, 2000*; *Maren, 2001*; *Pape and Pare, 2010*). Extinction of conditioned fear is an active learning process involving inhibition of fear expression. It is a decline in conditioned fear responses (CRs) following non-reinforced exposure to the CS. However, fear extinction memory does not erase the initial association between the CS and US but forms a new association (CS–No US) that inhibits expression of the previous conditioned memory (*Quirk and Mueller, 2008*). Fear extinction depends on specific structures such as the amygdala, the hippocampus, and the prefrontal cortex (PFC) (*Milad and Quirk, 2012*; *Orsini and Maren, 2012*). Dysfunctions in the neuronal circuits responsible for fear cause the development of anxiety disorders, including specific phobias and post-traumatic stress (*Pavlov, 1927*; *Quirk and Mueller, 2008*).

The consolidation of the extinction memory has been related to long-term synaptic plasticity and increases in the excitability of pyramidal neurons (PNs) from the infralimbic cortex (IL) (*Kaczorowski et al., 2012*; *Koppensteiner et al., 2019*; *Moyer et al., 1996*). The mechanisms of this synaptic plasticity involve NMDA receptors (*Burgos-Robles et al., 2007*), mitogen-activated protein (MAP) kinases (*Hugues et al., 2004*), protein kinase A (*Mueller et al., 2008*), insertion of $Ca^{2+}$-permeable AMPA receptors (*Sepulveda-Orengo et al., 2013*), and protein synthesis (*Mueller et al., 2008*; *Santini et al., 2004*). Moreover, the excitability of PN from IL is a key determinant for both the acquisition and the extinction of fear, being reduced by fear acquisition and increased by fear extinction (*Santini et al., 2008*). Indeed, IL layer 5 PNs (L5PNs) of conditioned fear animals show a higher slow post-spike afterhyperpolarization (sAHP) amplitude and lower firing frequency relative to non-conditioned or extinguished animals (*Santini et al., 2008*).

Insulin-like growth factor-1 (IGF-1) is an endogenous polypeptide with plenty of trophic functions, which can be locally synthesized and released by neurons and astrocytes (*Fernandez and Torres-Alemán, 2012*). Similarly, its receptor (IGF-1R) is widely expressed among all brain cell types (*Quesada et al., 2007*; *Rodriguez-Perez et al., 2016*). IGF-1 increases neuronal firing (*Gazit et al., 2016*; *Nuñez et al., 2003*) and modulates excitatory and inhibitory synaptic transmission in the central nervous system (*Castro-Alamancos and Torres-Aleman, 1993*; *Nilsson et al., 1988*; *Noriega-prieto et al., 2020*; *Seto et al., 2002*). Furthermore, IGF-1 regulates different ion channels, such as A-type $K^+$ channels (*Xing et al., 2006*) and P/Q-, L-, and N-type voltage-gated $Ca^{2+}$ channels (*Blair and Marshall, 1997*; *Shan et al., 2003*), as well as neurotransmitter receptors activity (*Gonzalez de la Vega et al., 2001*; *Savchenko et al., 2001*). However, the role of IGF-1 on the modulation of L5PN of the IL activity and its consequences in the fear extinction memory remain to be clarified.

Here we have examined the effects of IGF-1 in the excitability and synaptic transmission of L5PN from IL and in the extinction of conditioned fear. Our results reveal that IGF-1 induces a long-lasting reduction of the sAHP amplitude and increases the firing frequency of L5PNs. Furthermore, IGF-1 induces a presynaptic long-term depression (LTD) of both excitatory and inhibitory postsynaptic currents (EPSC and IPSCs, respectively) that results in a long-term potentiation (LTP) of the postsynaptic potentials (PSPs). Moreover, we show that IGF-1 facilitates the recall of extinction of fear conditioning when applied intracranially to the IL 30 min before the extinction task. In these animals with the favored extinction by IGF-1, the sAHP is reduced and the excitatory synaptic transmission is depressed. Therefore, we demonstrate for the first time that IGF-1 facilitates the extinction of fear conditioning, increasing the excitability and potentiating the synaptic transmission in L5PN from IL.

## Results

### IGF-1 increases the excitability of IL-L5PNs

$Ca^{2+}$-activated $K^+$ currents that mediate the medium and slow post-spike afterhyperpolarization (mAHP and sAHP, respectively) are crucial in the regulation of neuronal excitability (*Alger and Nicoll, 1980*; *Madison and Nicoll, 1984*). We first tested the effect of IGF-1 on different components of AHPs of IL L5PNs. In the current-clamp mode, we recorded mAHPs or sAHPs before and during bath application of IGF-1 (10 nM). IGF-1 reduced mAHP (*Figure 1A,B*) and sAHPs (*Figure 1A,C*), whereas the fast AHP (fAHP) was unaffected (*Figure 1—figure supplement 1A,B*). The reduction of mAHP and sAHP was abolished in the presence of the antagonist of the IGF-1 receptor, NVP-AEW 541 (*Figure 1A–C*). We also checked the effect on neuronal excitability by analyzing the number of action potentials (APs) evoked by increasing current injection steps. We observed that the number of APs evoked by this protocol was higher during IGF-1 perfusion (*Figure 1E,F*). Moreover, we compared the properties of AP before and after IGF-1 application. There were no significant differences in parameters, such as AP amplitude, AP threshold, AP half-width, AP time to max rise slope, AP max rise slope, AP time to max decay slope, and AP max decay slope (*Figure 1—figure supplement 2A–G*). However, the input resistance was significantly augmented (*Figure 1—figure supplement 2H*), suggesting a reduction of ionic conductance. We next performed voltage-clamp recordings to analyze the effect of IGF-1 on the currents that underlie the mAHP and sAHP (*Alger and Nicoll, 1980*; *Storm, 1990*) ($_mI_{AHP}$ and $_sI_{AHP}$, respectively). IGF-1 decreased $_mI_{AHP}$ (*Figure 2A–C*) and $_sI_{AHP}$ (*Figure 2A,D,E*), while the fast afterhyperpolarization current remained unaltered (*Figure 1—figure*

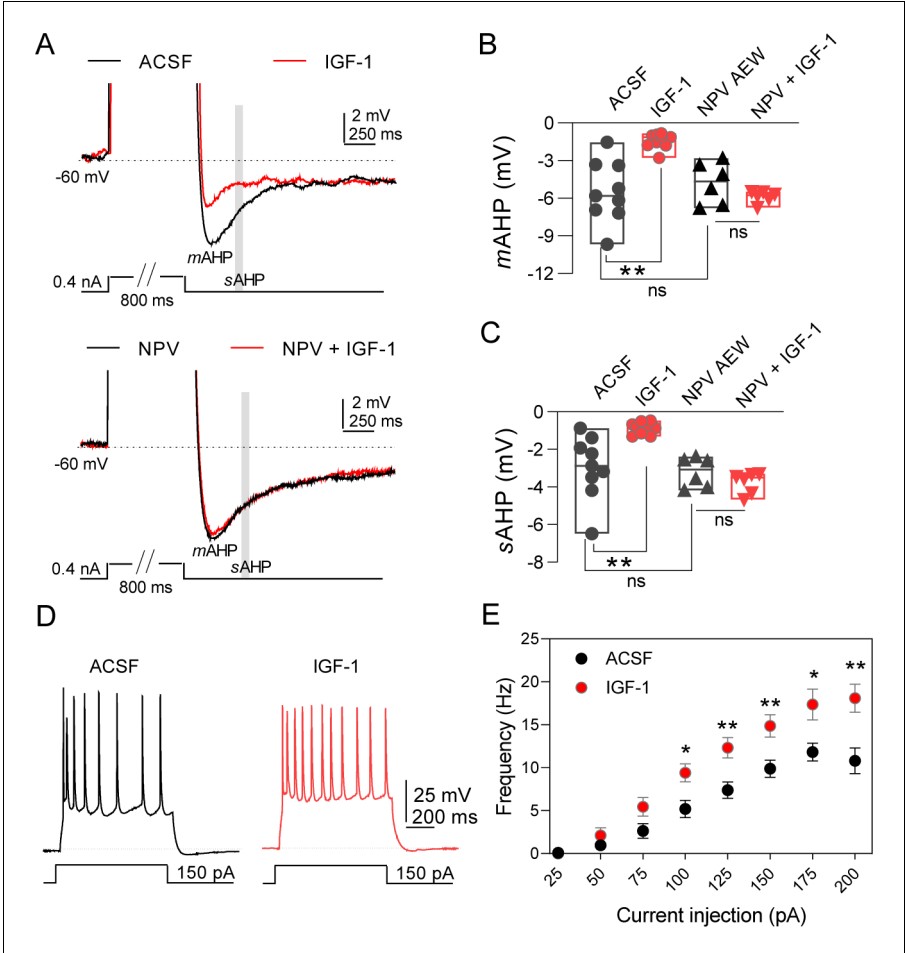

**Figure 1.** IGF-1 increases the excitability of IL-L5PNs. (**A**) Representative recordings from IL-L5PNs of hyperpolarizing potentials elicited by an 800 ms depolarizing pulse to study medium and slow AHP in control conditions (ACSF, top) and in the presence of NVP-AEW541 (40 nM, bottom), before (black) and during IGF-1 (10 nM, red) (spikes are truncated). The mAHP was measured at the hyperpolarization peak and the gray box indicates where the sAHP was measured. (**B**) Bar diagram summarizing mAHP amplitudes (n = 9 cells/8 animals; ACSF *vs* IGF-1 **p<0.01 and ns (non-significant) n = 6 cells/3 animals NVP *vs* NVP +IGF-1, Student's paired t-test). Mann–Whitney test, n = 9/6 cells ACSF *vs* NVP, ns. (**C**) Bar diagram summarizing sAHP amplitudes (n = 9 cells/8 animals; ACSF *vs* IGF-1 **p<0.01 and ns n = 6 cells/3 animals NVP *vs* NVP +IGF-1, Student's paired t-test). Mann–Whitney test, n = 9/6 cells ACSF *vs* NVP, ns. (**D**). Representative traces recorded from IL-L5PNs after 100 pA current injection in ACSF (gray) and after IGF-1 application (red). (**E**) Plot showing the frequency of APs as a function of the injected current (pA) for IL-L5PNs in ACSF (gray) and after IGF-1 application (red) (n = 10 cells/8 animals ACSF *vs* IGF-1 *p<0.05, **p<0.01 and ns, Multiple t-tests with post hoc Holm–Sildak multiple comparison methods). See also *Figure 1—figure supplement 1* and *Figure 1—figure supplement 2*.

The online version of this article includes the following source data and figure supplement(s) for figure 1:

**Source data 1.** Source data for *Figure 1*: IGF-1 increases the excitability of IL-L5PNs.

**Figure supplement 1.** IGF-1 has no effect on *f*AHP and *fl*AHP.

**Figure supplement 1—source data 1.** Source data for *Figure 1—figure supplement 1*.

**Figure supplement 2.** Action potential (AP) properties are unaltered by IGF-1.

**Figure supplement 2—source data 1.** Source data for *Figure 1—figure supplement 1*: Action potential properties are unaltered by IGF-1.

---

*supplement 1C,D*). We observed a gradual current reduction that reached a plateau 20 min after IGF-1 application in both the $_mI_{AHP}$ and $_sI_{AHP}$ (*Figure 2C,E*). However, in the absence of IGF-1, the neuronal depolarization did not modify the AHP currents (*Figure 2—figure supplement 1A*). Interestingly, we observed a long-lasting reduction of $_mI_{AHP}$ and $_sI_{AHP}$ that remained after the wash-out of

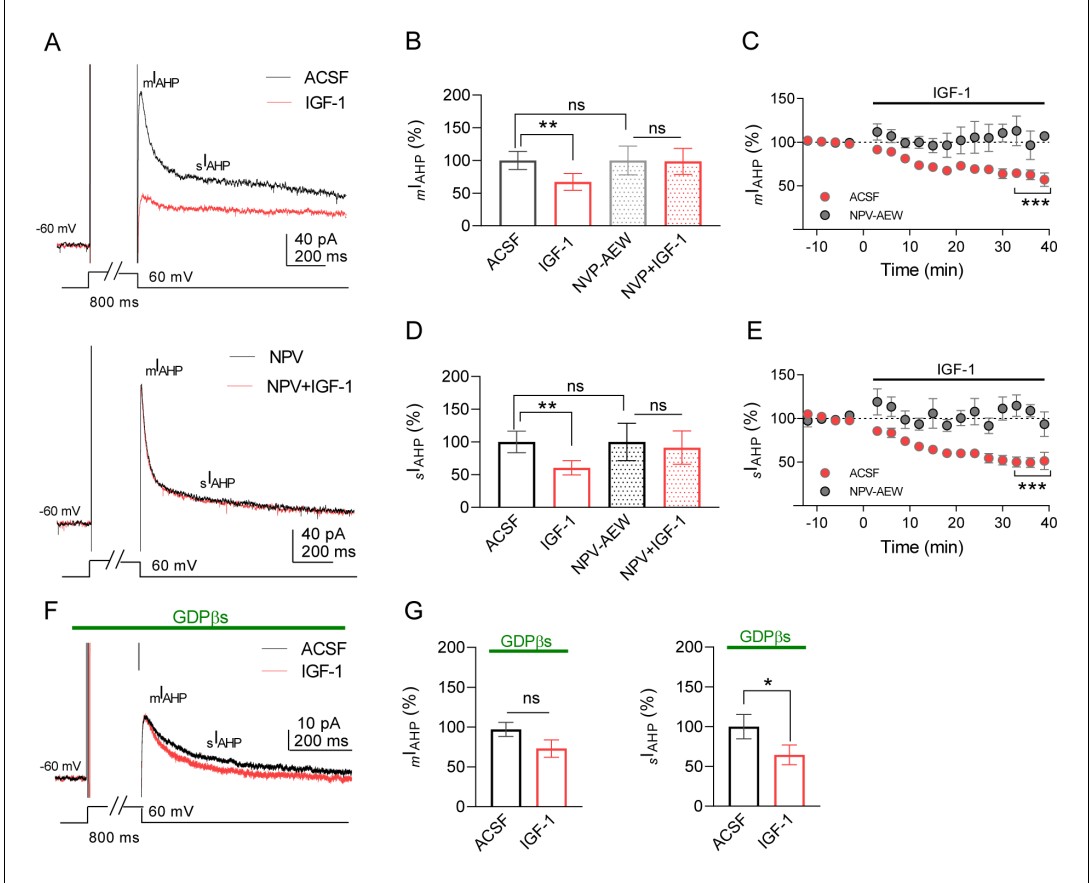

**Figure 2.** IGF-1 reduces $_mI_{AHP}$ and $_sI_{AHP}$ in IL-L5PNs. (A) Representative current traces recorded from IL-L5PNs in response to an 800 ms depolarizing voltage pulse from −60 mV to 0 mV in the control condition (ACSF, top) and the presence of NVP-AEW541 (40 nM, bottom), before (black), and during 10 nM IGF-1 (red). (B) Bar diagram summarizing the normalized amplitude of $_mI_{AHP}$ (n = 9 cells/5 animals; ACSF *vs* IGF-1 \*\*p<0.01 and ns, non-significant, n = 7 cells/7 animals NVP *vs* NVP + IGF-1, Student's paired t-test). Mann–Whitney test, n = 9/7 cells ACSF *vs* NVP, ns. (C) Plot showing the time course of the peak amplitude of the $_mI_{AHP}$ in ACSF (red) and in NVP-AEW541 before and during IGF-1 (n = 9 cells/7 animals; ACSF *vs* IGF-1 \*\*\*p<0.0001 and ns n = 7 cells/7 animals NVP *vs* NVP + IGF-1, Student's paired t-test). Mann–Whitney test, n = 9/7 cells IGF-1 *vs* NVP + IGF-1, ns. (D and E) Same as B and C respectively but for the $_sI_{AHP}$. (F) Same as A, but in the presence of GDPβs in the patch pipette. Note that only the $_sI_{AHP}$ is modulated by IGF-1 under blockade of the G proteins. (G) Bar diagram summarizing the normalized amplitude of $_mI_{AHP}$ (left) and normalized area of $_sI_{AHP}$ (right) under GDPβs (n = 8 cells/4 animals; $_mI_{AHP}$ ACSF *vs* IGF-1 ns, non-significant and $_sI_{AHP}$, n = 8 cells/4 animals ACSF *vs* IGF-1 \*p<0.05, Student's paired t-test). See also *Figure 2—figure supplement 1*.

The online version of this article includes the following source data and figure supplement(s) for figure 2:

**Source data 1.** Source data for *Figure 2*.

**Figure supplement 1.** Time course of $_mI_{AHP}$ and $_sI_{AHP}$.

**Figure supplement 1—source data 1.** Source data for *Figure 2—figure supplement 1*.

---

IGF-1 (*Figure 2—figure supplement 1B,C*) and required the activation of the AHPs by the depolarizing protocol (*Figure 2—figure supplement 1B,C*). Moreover, the IGF-1-dependent modulation of the $_mI_{AHP}$ and $_sI_{AHP}$ was prevented by the NVP-AEW541 (400 nM, *Figure 2A–E*), indicating that these effects were mediated by the activation of the IGF-1R.

We perform experiments to block the G protein-dependent modulation of the AHPs induced by metabotropic receptor activation. We used a patch pipette filled with the non-hydrolyzable GDP analogue GDPβs (1 mM) to block the G protein activation. Interestingly, we found that in the presence of GDPβs IGF-1 only reduced the $_sI_{AHP}$ (*Figure 2F,G*). It is noteworthy that although both AHPs were similarly reduced by IGF-1 in the same cells (32.8% for $_mI_{AHP}$ and 39.5% for $_sI_{AHP}$) the G protein independency of the $_mI_{AHP}$ modulation suggests that different mechanisms may be involved in the modulation of both AHPs. Taken together, these results demonstrate that IGF-1 induced a long-term decrease of $_mI_{AHP}$ and $_sI_{AHP}$ leading to an increase in the frequency of neuronal firing.

## IGF-1 induces LTP of the synaptic transmission

Next, we examined whether IGF-1 modulates the excitatory synaptic transmission. After isolating the EPSCs (see Materials and methods), we measured their amplitude and analyzed the paired-pulse ratio (PPR) and the coefficient of variance. IGF-1 induced a LTD of EPSC peak amplitude (72.63% of baseline, *Figure 3A*) that was prevented in the presence of NVP-AEW541 (97.72% of baseline, *Figure 3A*). The IGF-1-mediated depression of the EPSCs was associated with changes in the PPR (*Figure 3B*) and the coefficient of variation ($1/CV^2$) (*Figure 3C*), indicating a presynaptic mechanism. Additionally, this modulation was observed when the synaptic stimulation was absent (55.23% of baseline, *Figure 3D*), suggesting that the evoked synaptic responses were required for this LTD of the EPSCs.

We also analyzed whether IGF-1 regulates inhibitory synaptic transmission. IGF-1 induced a LTD of IPSCs (72.43% of baseline, *Figure 3E*) that were dependent on IGF-1R activation since this effect was prevented by NVP-AEW541 (99.0% of baseline, *Figure 3E*). The IGF-1-mediated depression of the IPSCs was associated with changes in the PPR (*Figure 3F*) and coefficient of variation ($1/CV^2$) (*Figure 3G*), pointing to a presynaptic mechanism. Nevertheless, the LTD of the IPSC did not require the evoked IPSCs since IGF-1 was able to induce it when the synaptic stimulation was absent (49.76% of baseline, *Figure 3H*). Interestingly, the IGF-1-dependent depression of EPSCs (*Figure 3A,D*) and IPSCs (*Figure 3E,H*) was larger in the absence of synaptic stimulation, suggesting that two different mechanisms may be responsible for the LTD. Finally, we studied the effect of IGF-1 on the PSPs. After recording a stable baseline of PSPs, we increased the intensity of stimulus until that ≈21% of the responses recorded during 5 min were suprathreshold and then we added IGF-1 (*Figure 3I,J*). IGF-1 induced a significant increase in the number of AP 10 and 15 min after its application (*Figure 3M*), whereas the properties of these AP were not modified (*Figure 3L,N*). After returning to the basal intensity of stimulation, a LTP of the PSPs was observed (*Figure 3I*). This PSP potentiation was not present when using the same protocol but in the absence of IGF-1 (*Figure 3—figure supplement 1*). According to the observed potentiation, IGF-1 application increased PSP amplitude, without modifying the PSP kinetics (*Figure 3J,K*). It is worth mentioning that a shorter period of IGF-1 application (10 min) was enough to induce an LTP of PSPs, which is maintained even after washing IGF-1 (*Figure 3—figure supplement 2*). Together, our results reveal that IGF-1 induces a presynaptic depression of both EPSCs and IPSCs which triggers an LTP of PSPs.

## IGF-1 facilitates fear extinction

Since a reduction of the sAHP favors the recall of extinction (*Santini et al., 2008*), we next analyzed whether IGF-1 was able to improve it. First, we performed a set of experiments to determine the number of sessions required to induced fear conditioning and extinction as previously described (*Fontanez-Nuin et al., 2011*; *Figure 4—figure supplement 1*). On day 2, the different groups of rats acquired similar levels of conditioned freezing (COND, 74%; EXT, 77%) with three sessions of association (CS-US, fear conditioning). On day 3, rats from the EXT group showed gradual within-session extinction across 20 trials to a final freezing level of 12% (CS-No US), while rats from the COND group remained in their home cage. On day 4, rats from COND group showed high levels of freezing (77–95%) on the test tones, whereas rats from EXT group showed low levels of freezing (50–35%), indicating good recall of extinction during five sessions (*Figure 4—figure supplement 1A,B*).

To study the effect of IGF-1 on fear extinction memory, we infused saline or IGF-1 or NVP + IGF-1 through a cannula guide previously implanted into the IL (*Figure 4A*) 30 min before the extinction training (*Figure 4B*). Although the level of freezing reached on day 2 was comparable in the three groups (saline, 89%; IGF-1, 79%; NVP + IGF-1 81%), the rats from the IGF-1 group showed less freezing in the last session on day 3 (saline, 51%; IGF-1, 14%; NVP + IGF-1, 59%), indicating that IGF-1 treatment favors the acquisition of extinction. On day 4, IGF-1-treated rats exhibited less freezing (37–19%) than the saline (57–41%) and NVP + IGF-1 (63–59%) rats (*Figure 4B*), unveiling that IGF-1 enhanced the performance of animals to recall of extinction.

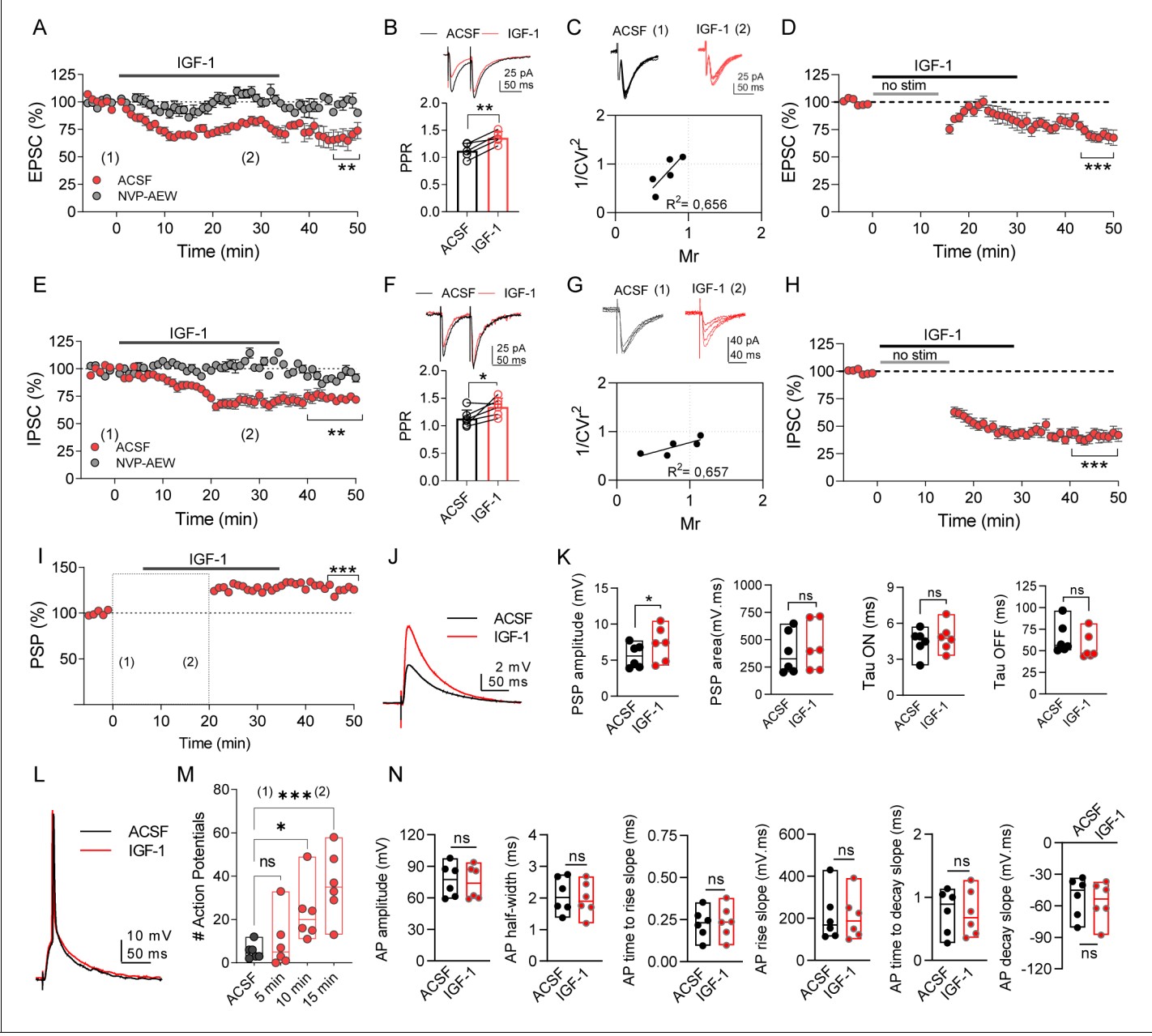

**Figure 3.** IGF-1 induces long-term potentiation of the synaptic transmission. (**A**) Plot showing the time course of EPSCs when the IGF-1 was bath applied for 35 min after a stable (~5 min) baseline, (n = 5 cells/3 animals, ACSF *vs* IGF-1 ***p<0.001 and ns, n = 5 cells/3 animals NVP *vs* NVP +IGF-1 Student's paired t-test). Mann–Whitney test, n = 5/5 cells IGF-1 *vs* NVP +IGF-1, **p<0.01. (**B**) (top) Superimposed representative paired-pulse EPSCs recorded before (black trace) and during IGF-1 (red trace). (bottom) Bar diagram summarizing the EPSC paired-pulse ratio (PPR) before (black bar) and during IGF-1 (red bar). Student's paired t-test **p<0.01; (n = 5, same as **A**). (**C**) (top) Representative EPSCs traces before and during IGF-1. (bottom) Plot of the variance (1/CV$^2$r) as a function of the mean EPSC peak amplitude (**M**) 30 min after IGF-1 and normalized to control condition (ACSF) (n = 5, same as **A**). (**D**) Plot showing the time course of EPSCs when IGF-1 is applied in the absence of synaptic stimulus for 15 min. (n = 5 cells/3 animals, ACSF *vs* IGF-1 ***p<0.001 Student's paired t-test). (**E–H**) Same as **A–D**, respectively, but for the IPSCs. (**I**) Plot showing the time course of PSP amplitude when, after a stable (~5 min) baseline, IGF-1 was bath applied for 35 min. Student's paired t-test ***p<0.001 (n = 6). (**J**) Representative PSPs recordings before (black trace) and during IGF-1 (red trace). (**K**) Bar diagrams summarizing the effect of IGF-1 on the PSP amplitude, PSP area, and the TAU ON and OFF (n = 6 cells/4 animals, ACSF *vs* IGF-1 *p<0.05 Student's paired t-test). Note that only the amplitude and area of the PSPs are modulated by IGF-1. (**L**) Representative current-clamp responses recorded before (black trace) and during IGF-1 (red trace). (**M**) Bar diagram summarizing the number of action potentials (APs) in ACSF and at 5, 10, and 15 min during IGF-1 (n = 6 cells/4 animals, ACSF *vs* IGF-1 *p<0.05, ***p<0.001 one-way ANOVA with post hoc Dunnett's multiple comparison test). (**N**) Bar diagram summarizing the lack of effect of IGF-1 in the AP

*Figure 3 continued on next page*

*Figure 3 continued*

amplitude, the AP half-width, AP time to rise slope, AP rise slope, AP time to decay slope, and AP decay slope (n = 6 cells/4 animals, ACSF vs IGF-1 Student's paired t-test). See also *Figure 3—figure supplement 1* and *Figure 3—figure supplement 2*.

The online version of this article includes the following source data and figure supplement(s) for figure 3:

**Source data 1.** Source data for *Figure 3*.
**Figure supplement 1.** The protocol of stimulation does not induce plasticity.
**Figure supplement 1—source data 1.** Source data for *Figure 3—figure supplement 1*.
**Figure supplement 2.** The application of IGF-1 is sufficient to induce plasticity.
**Figure supplement 2—source data 1.** Source data for *Figure 3—figure supplement 2*.

## IGF-1 modulates excitability and synaptic transmission in rats exposed to the extinction memory task

Fear extinction is paralleled by an increase in the excitability of PNs from layer 5 from IL (*Santini et al., 2008*). Therefore, we tested whether a change in the excitability of IL-L5PNs occurs in the animals in which IGF-1 facilitated the extinction memory. For this purpose, we recorded IL-L5PNs from all behavioral groups after the last extinction session and we observed that the $_mI_{AHP}$ and $_sI_{AHP}$ of the IGF-1 group were smaller than the $_mI_{AHP}$ and $_sI_{AHP}$ of the saline group (*Figure 4C–E*). Importantly, these differences in the $_mI_{AHP}$ and $_sI_{AHP}$ were not present when comparing the saline group with the group treated with NVP + IGF-1 (*Figure 4C–E*). Also, we calculated the correlation between the $_mI_{AHP}$ and $_sI_{AHP}$ amplitudes and freezing levels in all groups. Interestingly, both current amplitude correlated with the freezing (*Figure 4F*), indicating that the reduction in the currents mediated by IGF-1 facilitates the extinction recall. It is noteworthy that all rat groups that underwent the extinction, except for the Ext-NVP +IGF-1 group, showed a reduced fAHP and mAHP when compared with NAÏVE group (*Figure 4—figure supplement 2A,B*), which is in agreement with previous results (*Santini et al., 2008*). However, we only observed a significant reduction in sAHP from Ext-IGF-1 group compared NAÏVE group, suggesting that in this group of animals the reduction of the sAHP is essential for the better level of extinction recall (*Figure 4—figure supplement 2C*). Moreover, IGF-1-treated group showed an increased AP firing frequency of IL-L5PNs when compared with saline or NVP + IGF-1 groups. These results support that IGF-1 induces the facilitation of extinction through the reduction of $_mI_{AHP}$ and $_sI_{AHP}$ thus increasing the IL-L5PNs firing frequency (*Figure 4G*). Finally, we also tested whether the excitatory synaptic transmission was depressed by IGF-1 in animals in which IGF-1 facilitated the extinction. For this purpose, we recorded miniature EPSCs (mEPSCs) in IL-L5PNs from all behavioral groups after the last extinction session on day 4. We observed that the mEPSCs frequency of the IGF-1 group was lower than in saline and NVP + IGF-1 groups (*Figure 5A–C*), with no changes in the mEPSCs amplitude (*Figure 5A,D,E*) suggesting a presynaptic modulation of synaptic transmission by IGF-1. Taken together, these results demonstrate that IGF-1 facilitates the establishment of fear extinction memory, generating plasticity of synaptic transmission and neuronal excitability at IL-L5PNs.

## Discussion

Interactions among the amygdala, the hippocampus, and the IL PFC are important for the extinction of conditioned fear memory (*Milad and Quirk, 2012*; *Orsini and Maren, 2012*; *Pape and Pare, 2010*; *Quirk and Mueller, 2008*). Although the effects of IGF-1 on the amygdala (*Stern et al., 2014*) and the hippocampus (*Chen et al., 2011*; *Stern et al., 2014*) have been previously studied, there is no evidence about its possible actions on the IL. Our results demonstrate for the first time that IGF-1 applied to IL favors the extinction of conditioned fear memory causing an increase in L5PN excitability and synaptic plasticity through the activation of the IGF-1R. We present new evidence showing that IGF-1 induced a long-lasting increase in excitability and LTP of the synaptic potentials at IL-L5PNs. The former is mediated by a significant reduction in the mAHP and sAHP, whereas the latter results from the interaction between the presynaptic LTD of the EPSCs and IPSCs. Therefore, our results show a novel functional consequence of IGF-1 signaling on animal behavior in the mPFC through the modulation of the extinction of conditioned fear memory.

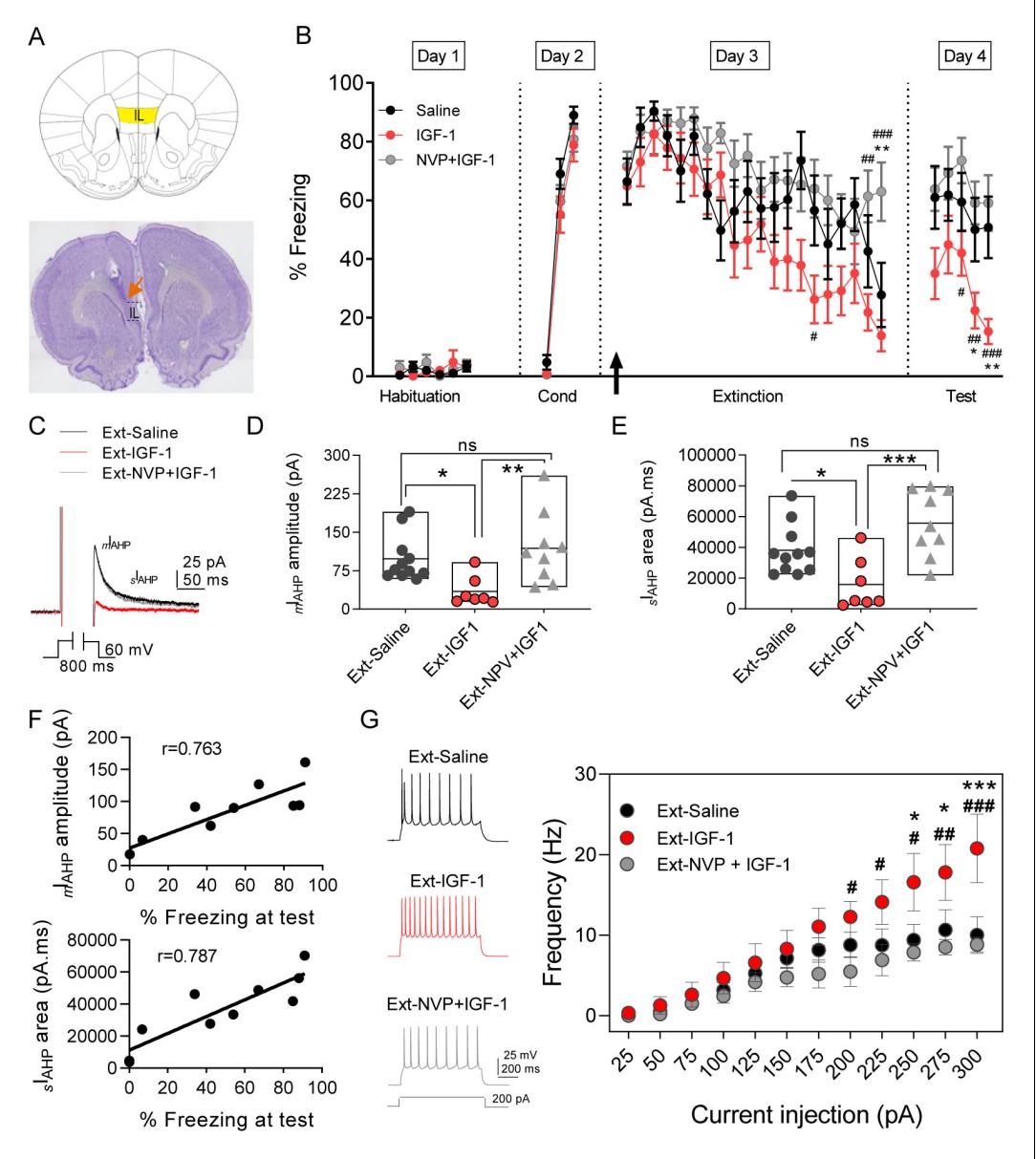

**Figure 4.** IGF-1 facilitates the extinction memory by reducing $_mI_{AHP}$ and $_sI_{AHP}$. (**A**) (top) Image showing the infralimbic cortex (IL) localization. (bottom) Nissl-stained coronal section of rat IL, the orange arrow indicates the tip of the cannula implanted on the IL. (**B**) Plot showing the time course of the percentage of freezing during the extinction protocol for the three groups studied (saline, IGF-1, and NVP-+IGF-1). Saline (n = 11), IGF-1 (n = 14), and NVP-AEW541 +IGF-1 (n = 12) were directly applied into the IL through the implanted cannula 30 min before extinction training (day 3). The black arrow indicates the time of infusion. (*p<0.05 **p<0.01 saline *vs* IGF-1 and NPV + IGF-1 #p<0.05; ##p<0.01; ###p<0.001 IGF-1 *vs* NPV + IGF-1; One-way ANOVA with post hoc Tukey's multiple comparisons test). (**C**) Representative mI_AHP and sI_AHP current traces recorded in three groups of animals studied that showed fear extinction (**D**) Bar diagrams summarizing the $_mI_{AHP}$ for all the groups studied. Recordings were performed in day 4 (test) after finishing the extinction protocol; saline (n = 11 cells/3 animals) *vs* IGF-1 (n = 7 cells/4 animals) *p<0.05; IGF-1 *vs* NPV + IGF-1 (n = 9 cells/3 animals) **p<0.01 and saline (n = 11 cells/3 animals) *vs* NPV + IGF-1 (n = 9 cells/3 animals) ns, one-way ANOVA with post hoc Tukey's multiple comparisons test. (**E**) Bar diagrams summarizing the $_sI_{AHP}$ for all the groups studied. Saline (n = 11 cells/3 animals) *vs* IGF-1 (n = 7 cells/4 animals) *p<0.05; IGF-1 vs NPV + IGF-1 (n = 9 cells/3 animals) **p<0.01 and saline (n = 11 cells/3 animals) *vs* NPV + IGF-1 (n = 9 cells/3 animals) ns, one-way ANOVA with post hoc Tukey's multiple comparisons test. (**F**) For each rat in the extinction groups, the $_mI_{AHP}$ and $_sI_{AHP}$ for all cells were averaged. Across rats, these values were significantly correlated with freezing levels at test (r = 0.763 for $_mI_{AHP}$ and r = 0.787 for $_sI_{AHP}$ pointing to that the

*Figure 4 continued on next page*

*Figure 4 continued*

reduction in both currents is correlated with the extinction recall). (G) Representative traces recorded from IL-L5PNs from animals that showed fear extinction after 200 pA (left) current injection in saline (black); IGF-1 (red) and NVP + IGF-1 (gray). Frequency-injected current relationships for IL-L5PNs in saline (black, n = 10 cells/5 animals) IGF-1 (red, n = 7 cells/3 animals) and NVP + IGF-1 (gray, n = 6 cells/3 animals). (*) IGF-1 *vs* saline and (#) IGF-1 *vs.* NVP + IGF-1; one-way ANOVA with post hoc Tukey's multiple comparisons test. See also *Figure 4—figure supplement 1* and *Figure 4—figure supplement 2*.

The online version of this article includes the following source data and figure supplement(s) for figure 4:

**Source data 1.** Source data for *Figure 4*.
**Figure supplement 1.** Protocol of behavior.
**Figure supplement 1—source data 1.** Source data for *Figure 4—figure supplement 1*.
**Figure supplement 2.** AHPs of the L5PN of the IL recorded from animals after fear extinction.
**Figure supplement 2—source data 1.** Source data for *Figure 4—figure supplement 2*.

## IGF-1-induces AHPs reduction leading to enhancement of excitability

The enhancement of excitability is manifested as a reduced spike frequency adaptation in response to prolonged depolarizing current applications, which results from post-burst AHP reduction (*Moyer et al., 1996*; *Sepulveda-Orengo et al., 2013*) modulation of intrinsic excitability is linked with the mechanism of consolidation of learning (*Sehgal et al., 2014*). Specifically, in IL, a reduction of excitability maintains the fear learning (*Criado-Marrero et al., 2014*) whereas an increase of the excitability regulated the fear extinction learning (*Sepulveda-Orengo et al., 2013*; *Bloodgood et al., 2018*). Interestingly, our recordings from IL-L5PNs of animals treated with IGF-1 showed a long-lasting decrease of mAHP and sAHP currents, which is highly related to behavioral recall (*Moyer et al., 1996*; *Santini et al., 2008*), suggesting that IGF-1 also modulates the channels responsible for these currents in IL-L5PNs. A previous study demonstrated that the enhancement of the bursting of these neurons after the acquisition of fear extinction is mediated by a reduction of $_{fl}I_{AHP}$ due to the blockade of the M-type K$^+$ current (*Santini and Porter, 2010*). The reduction in the amplitude and the duration of AHPs from IL-L5PNs caused by IGF-1 resulted in a significant increase in their excitability, as reflected in the increase of their firing frequency. In the present work, we demonstrate that IGF-1 does not modify the $_{fl}I_{AHP}$, suggesting that the effects of IGF-1 on IL-L5PN firing frequency and recall of extinction are not mediated by the modulation of the M-current. Moreover, the AP characteristics are not affected by IGF-1, suggesting that the IGF-1-mediated increase in IL-L5PNs excitability is mainly produced by the decrease in the mAHP and sAHP currents. Although IGF-1 inhibits A-type K$^+$ channels in somatosensory neurons in the dorsal column nuclei (*Blair and Marshall, 1997*; *Nuñez et al., 2003*; *Shan et al., 2003*; *Xing et al., 2006*) this modulation is absent in the IL.

The activation of metabotropic glutamate receptor 5 (mGluR5) produces a reduction of $_m I_{AHP}$ and $_s I_{AHP}$, an effect that is prevented in the presence of a mGluR5 antagonist (*Fontanez-Nuin et al., 2011*; *Young et al., 2008*), pointing to mGluR5 as responsible for the modulation of these conductances. In our study, inhibition of G protein-coupled receptors by GDPβs abolished the IGF-1-mediated reduction of mIAHP, suggesting that the effect of IGF-1 on mAHP is mediated by the activation of metabotropic receptors. One candidate is the mGLuRs, since it has been demonstrated that the activation of mGluR5 increases IL-L5PN excitability (*Fontanez-Nuin et al., 2011*) and modulates the recall of extinction (*Fontanez-Nuin et al., 2011*; *Sepulveda-Orengo et al., 2013*) in a similar way to what we describe for IGF-1. However, we cannot rule out the involvement of other GPCRs and a further study is required to analyze the type of metabotropic receptor and the signaling pathway mediating the reduction of the mAHP by the IGF-1. On the other hand, we observed that IGF-1-mediated sAHP reduction is independent of the activation of G-protein-coupled receptors, which suggests that IGF-1 could be directly modulating the activity of some AHP-related ion channels, such as SK channels.

## IGF-1-induces an LTP of the PSPs

The inhibitory transmission controls the operation of cortical circuits and the modulation of synaptic inhibition plays an important role in the induction of cortical plasticity. In response to glutamate,

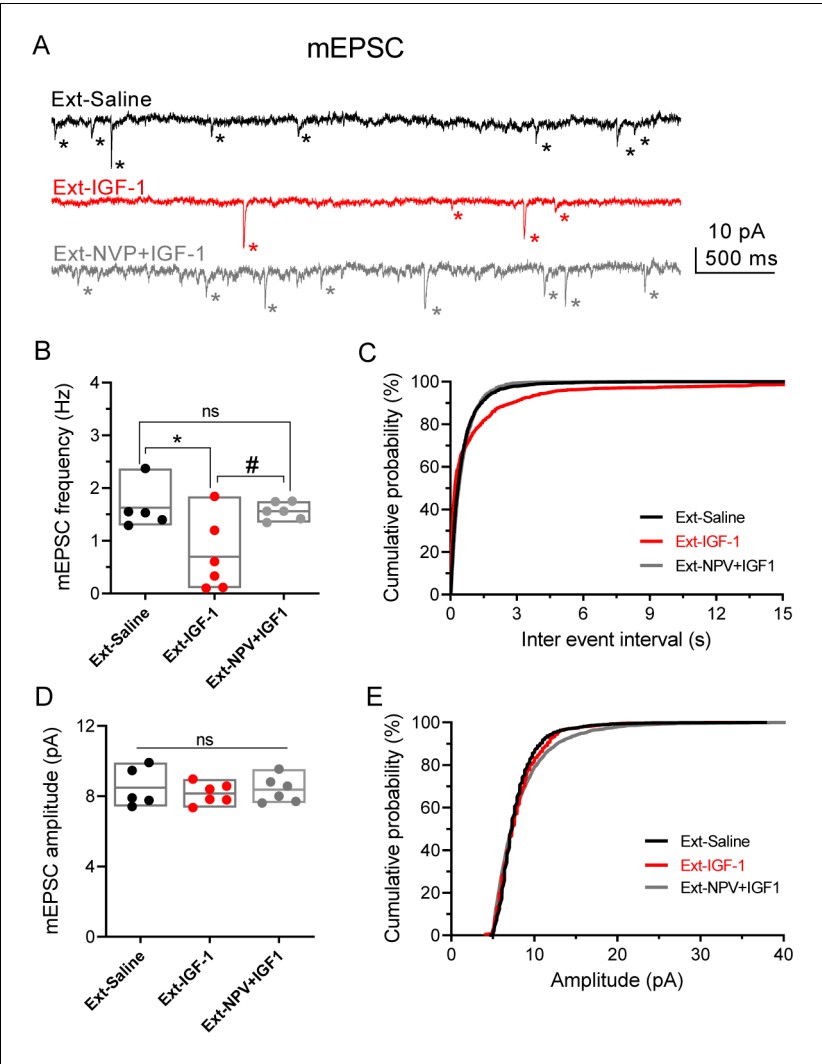

**Figure 5.** IGF-1 decreases the frequency of mEPSC. (A) Representative traces recorded at −60 mV in IL-L5PNs from animals that showed fear extinction, in the presence of 1 µM TTX, 50 µM PiTX, and 5 µM CPG-55845. Asterisks denote mEPSC events. Note the decreased mEPSCs frequency induced by IGF-1. (B) Bar diagram of the summary data showing mean mEPSCs frequency from Ext-Saline (black, n = 5 cells/2 animals); Ext-IGF-1 (red, n = 6/3 animals); and Ext-NVP +IGF-1 (gray, n = 6/3 animals). One-way ANOVA with post hoc Tukey's multiple comparisons test. (C) Cumulative probability plots of mean inter-mEPSC interval in Ext-Saline (black); Ext-IGF-1 (red); and Ext-NVP +IGF-1 (gray) (same cells as in B). Note that IGF-1 increased the mean inter-mEPSC interval (p<0.05; Kolmogorov–Smirnov test). (D) Bar diagram of the summary data showing mean mEPSCs amplitude (same cells as in B). (E) Cumulative probability plots of mean amplitude-mEPSC in Ext-Saline (black); Ext-IGF-1 (red); and Ext-NVP +IGF-1 (gray). Note that IGF-1 does not change the mean mEPSCs amplitude (same cells as in B).

The online version of this article includes the following source data for figure 5:

**Source data 1.** Source data for *Figure 5*: IGF-1 decreases the frequency of mEPSC.

IGF-1 induces a long-lasting reduction in GABA release from cerebellar Purkinje cells, suggesting that IGF-1 is modulating the glutamatergic transmission in the rat olivo-cerebellar system (*Castro-Alamancos and Torres-Aleman, 1993*; *Castro-Alamancos et al., 1996*). IGF-1 was also shown to modulate GABAergic transmission in the olfactory bulb (*Liu et al., 2017*). However, this is the first demonstration of an IGF-1-mediated presynaptic long-lasting depression of fast GABAergic synaptic transmission in the PFC. Although IGF-1Rs may be present in the inhibitory GABAergic terminals, there are no clear data showing a direct action of IGF-1 acting on presynaptic IGF-1Rs. The only

evidence suggesting this mode of action was seen in the hippocampus, where IGF-1 can induce the release of GABA to regulate endogenous ACh release, possibly acting via GABAergic neurons (*Seto et al., 2002*).

It has been demonstrated that excitatory synaptic transmission is also modulated by IGF-1 in many brain areas (*Araujo et al., 1989*; *Castro-Alamancos and Torres-Aleman, 1993*; *Maya-Vetencourt et al., 2012*; *Nilsson et al., 1988*; *Seto et al., 2002*). IGF-1 enhances glutamatergic synaptic transmission through a postsynaptic mechanism involving postsynaptic AMPA, but not NMDA, receptors in CA1 PNs from rat hippocampus (*Molina et al., 2012*; *Ramsey et al., 2005*). Additionally, IGF-1 increases the expression levels of hippocampal NMDA receptor subunits 2A and 2B in aged rats (*Sonntag et al., 2000*), which may facilitate LTP induction. However, here we show that IGF-1 decreases the efficacy of excitatory synaptic transmission by a presynaptic mechanism. Therefore, our observations contribute to the notion that IGF-1 can decrease the release of neurotransmitters both at excitatory and inhibitory synapses throughout the CNS. The net effect of this modulation by IGF-1 is a long-lasting potentiation of the PSPs indicating a greater impact of the reduction of inhibition compared with the reduction of excitation. Interestingly, we observed that two mechanisms coexist in the development of the LTD of EPSCs and IPSCs following IGF-1 application. The first one is dependent on synaptic activation and is characterized by a fast onset, the plateau being achieved at 10–15 min after IGF-1 application. The second one is independent of synaptic activation and is characterized by a slower onset, achieving the plateau at 40 min after IGF-1 application. However, the increase in the input resistance induced by IGF-1 could also explain the resultant LTP of the PSPs nonetheless of the reduction in the release of neurotransmitters. Since SK channels have been shown to modulate NMDA receptors (*Jones et al., 2017*; *Ngo-Anh et al., 2005*), we cannot discard a putative IGF-1-dependent modulation of SK channels as the source of the potentiation in the synaptic transmission. The resultant LTP of the PSPs and the increase in the input resistance summed to the decrease of the AHPs would result in the increase of the L5PN activity of the IL and its effect on the fear extinction memory.

## IGF-1 facilitates fear extinction memory

There is a growing body of evidence supporting that fear extinction memory is encoded by IL neurons (*Quirk and Beer, 2006*) that show enhanced responses to extinguished cues during extinction recall (*Milad and Quirk, 2002*). Pharmacological (*Hugues et al., 2004*; *Laurent and Westbrook, 2009*; *Sierra-Mercado et al., 2011*; *Sotres-Bayon et al., 2007*), electrical (*Milad et al., 2004*), or optogenetic (*Do-Monte et al., 2015*) manipulations of IL have been described to modulate the acquisition of fear extinction. In addition, IL is involved in the recall of this memory since lesions of the IL produce deficits in its retention (*Morgan and LeDoux, 1995*; *Quirk et al., 2000*). In the present study, we infused IGF-1 into IL before the extinction protocol and found that the action of IGF-1 in these neurons induced a significant improvement of fear extinction memory compared to the groups treated with saline or NVP + IGF-1. Consistent with that fact, IGF-1 (*Llorens-Martín et al., 2010*; *Trejo et al., 2008*), and most recently also IGF-2 (*Chen et al., 2011*), have been related to cognitive function. Injections of IGF-1, IGF-2, or insulin into the amygdala did not affect memories engaging this region. However, bilateral injection of insulin into the dorsal hippocampus transiently enhances hippocampal-dependent memory, whereas injection of IGF-1 has no effect (*Stern et al., 2014*). IGF-2 produces the most potent and persistent effect as a memory enhancer on hippocampal-dependent memories (*Chen et al., 2011*). Like insulin, IGF-2 did not affect amygdala-dependent memories when delivered into the BLA. Contextual fear extinction is facilitated when IGF-2 is injected into the dentate gyrus whereas inhibition of physiological IGF-1 signaling, via intrahippocampal injection of an IGF-1 blocking antibody, did not affect fear extinction (*Chen et al., 2011*). However, a single intravenous injection of IGF-1 before the training of contextual fear extinction increases the density of mature dendritic spines in the hippocampus and mPFC, favoring the memory of extinction (*Burgdorf et al., 2017*). All these reports are consistent with the idea that IGF-2 would have a specific role in the hippocampus while IGF-1 could have a selective role in the IL.

The reduction in the AHP amplitude associated with learning mechanisms is transitory since AHP amplitude initial values are recovered within days when training is finished, indicating an intrinsic plasticity phenomenon (*Moyer et al., 1996*). Cohen-Matslish and coworkers showed that the AHP reduction induced by the synaptic activation requires protein synthesis during a specific time-window for its long-term maintenance (*Cohen-Matsliah et al., 2010*). It has been shown that the acquisition

of extinction memory depends on protein synthesis (*Santini et al., 2004*). In our study, we show that IGF-1 favors the acquisition of extinction memory when applied directly to the IL in vivo. Therefore, it is tempting to hypothesize that the slow onset kinetics of AHP reduction that we observed after bath perfusion of IGF-1 in brain slices are suggestive of a mechanism involving protein synthesis, although further experiments would be needed to confirm it. Consistent with our observations in brain slices (plateau of AHP reduction is reached after 30 min), AHP amplitude is reduced in IL neurons from rats where the extinction memory was consolidated (recordings from the test day in IGF-1-treated animals). Moreover, we found that a reduction in AHP amplitude was directly correlated with a reduction in the expression of fear, suggesting that mechanisms favoring a greater AHP reduction would produce an enhancement of the acquisition of extinction memory.

Following the acquisition of fear extinction memory, there is a preferential shift toward increased intrinsic excitability in IL-amygdala communication (*Bloodgood et al., 2018*). In our study, we correlate the reduction in AHP amplitude with an increase in IL neuronal excitability. Under these circumstances, this increase in IL neurons' bursting would activate inhibitory intercalated cells in the amygdala to produce the inhibition of fear expression, as described previously (*Quirk and Mueller, 2008*).

### Functional relevance of the IGF-1 modulation in the mPFC

IGF-1 is actively transported to the central nervous system from plasma through the choroid plexus (*Carro et al., 2000*), and it is also locally produced in the brain by neurons and glial cells (*Quesada et al., 2007*; *Suh et al., 2013*; *Rodriguez-Perez et al., 2016*). Interestingly, the uptake of IGF-1 by the brain correlates with frequency-dependent changes in cerebral blood flow in the cortex during information processing (*Nishijima et al., 2010*). Therefore, the levels of IGF-1 in the IL would depend on both its active transport from the plasma, favored by the activity involved in information processing, and in the IGF-1 locally produced by neurons and astrocytes of the IL. Thus, high levels of IGF-1 are expected in the IL because of its high activity during the extinction of fear conditioning (*Milad and Quirk, 2012*; *Sepulveda-Orengo et al., 2013*). Our results demonstrate that a further increase in the IGF-1 levels due to the exogenous application of IGF-1 in the IL favor the extinction of conditioned fear. In natural conditions, these high levels of IGF-1 could be expected after physical exercises because it is known that IGF-1 uptake from the plasma and IGF-1 levels in the brain are increased after running, reaching the higher levels (*Carro et al., 2000*). Thus, our results could be related to some of the benefits of physical exercise on the brain function by increasing neuronal excitability and favoring synaptic plasticity and learning and memory through the activation of the IGF-1Rs.

In conclusion, the present findings reveal novel mechanisms and functional consequences of IGF-1 signaling in IL. On one hand, IGF-1 induces a reduction in $mI_{AHP}$ and $sI_{AHP}$ and increases the excitability of L5PNs of IL. On the other hand, IGF-1 modulates the excitatory and inhibitory synaptic transmission resulting in a long-lasting enhancement of the synaptic efficacy. Both the synaptic and intrinsic plasticity regulate the neuronal connectivity, which leads to the facilitation of consolidation of fear extinction memory. Altogether, these results strongly support the potential role of IGF-1 as a new therapeutic target for the treatment of anxiety and mood disorders.

## Materials and methods

### Animals

Male Sprague Dawley rats were group-housed in transparent polyethylene cages. Rats were maintained on a 12:12 hr light/dark scheduled cycle with free access to food and water. All animal procedures were approved by the Universidad Autónoma of Madrid Ethical Committee on Animal Welfare and conform to Spanish and European guidelines for the protection of experimental animals (Directive 2010/63/EU). An effort was made to minimize animal suffering and number.

### Electrophysiology

Prefrontal cortical slices were obtained from rats at postnatal day (P20–P30) age. Rats were decapitated and the brain removed and submerged in artificial cerebrospinal fluid (ACSF). Coronal slices (400 µm thick) were obtained with a Vibratome (Leica VT 1200S). To reduce swelling and damage in

superficial layers (especially after the behavior tests), brain slices were obtained using a modified ACSF, containing (in mM): 75 NaCl, 2.69 KCl, 1.25 $KH_2PO_4$, 2 $MgSO_4$, 26 $NaHCO_3$, 2 $CaCl_2$, 10 glucose, 100 sucrose, one sodium ascorbate, and three sodium pyruvate. After that, brain slices were transferred to regular ACSF (in mM: 124 NaCl, 2.69 KCl, 1.25 $KH_2PO_4$, 2 $Mg_2SO_4$, 26 $NaHCO_3$, 2 $CaCl_2$, and 10 glucose, 0.4 sodium ascorbate, bubbled with carbogen [95% $O_2$, 5% $CO_2$]) and incubated for >1 hr at room temperature (22–24°C). Slices were then transferred to an immersion recording chamber and superfused with carbogen-bubbled ACSF. Cells were visualized under an Olympus BX50WI microscope. Patch-clamp recordings were obtained from the soma of PNs located at the layer 5 of IL (IL-L5PNs) using patch pipettes (4–8 MΩ) filled with an internal solution that contained (in mM): 130 $KMeSO_4$, 10 HEPES-K, 4 $Na_2ATP$, 0.3 $Na_3GTP$, 0.2 EGTA, 10 KCl (buffered to pH 7.2–7.3 with KOH). Recordings were performed in current- or voltage-clamp modes using a Cornerstone PC-ONE amplifier (Dagan Corporation). Pipettes were placed with a micromanipulator (Narishige). The holding potential was adjusted to −60 mV and the series resistance was compensated to ~80%. Recordings were accepted only when series and input resistances did not change >20% during the experiment. The liquid junction potential was not compensated. Data were low-pass filtered at 3 kHz and sampled at 10 kHz, through a Digidata 1440 (Molecular Devices). The pClamp software (Molecular Devices) was used to acquire the data. Chemicals were purchased from Sigma-Aldrich Quimica and Tocris Bioscience (Ellisville; distributed by Biogen Cientifica) and R and D Systems, Inc (distributed by Bio-Techne). In the current-clamp mode, the afterhyperpolarizing potentials (AHPs) were studied using two different pulse protocols. The fast AHP (fAHP) was evoked by a 10 ms depolarizing pulse whereas an 800 ms depolarizing pulse was used to generate the medium and the slow AHP (mAHP and sAHP, respectively). While the fAHP and mAHP were measured as peak amplitude of the afterhyperpolarization, the sAHP was estimated as the average membrane potential between 50 ms and 280 ms after the 800 ms pulse (*Santini et al., 2008*). We applied the same protocol but in the voltage-clamp mode to record the currents that underline these potentials. We fixed the membrane potential at −60 mV and then we depolarized the cell at 0 mV during 10 ms and measured the $_fI_{AHP}$ at the peak of the currents. In the case of $_mI_{AHP}$ and $_sI_{AHP}$ we depolarized the cell at 0 mV during 800 ms. We measured the $_mI_{AHP}$ at the peak of the current and measured the $_sI_{AHP}$ as the area under the curve during 1 s period beginning at the peak of the current. We also analyzed the excitability of IL-L5PNs by measuring the number of APs (spike frequency) elicited in response to a series of long (1 s) depolarizing current steps (25–200 pA for basal condition and 25–350 pA for neurons recorded after the behavioral tests; 25 pA increments) before and during IGF-1. The AP voltage threshold was defined as the first point on the rising phase of the spike with a change in voltage exceeded 50 mV/ms. The spike amplitude was quantified as the difference between the AP voltage threshold and the peak voltage. The spike width was measured at 1/2 of the total spike amplitude. The waveform characteristics of the APs recorded from IL-PNL5s, i.e., rise time, rise slope, decay time, and decay slope, were determined using Clampfit10 software (Axon Instruments).

Bipolar stimulation was applied through a Pt/Ir concentric electrode (OP: 200 μm, IP: 50 μm; FHC) connected by two silver-chloride wires to a stimulator and a stimulus isolation unit (ISU-165 Cibertec). The stimulating electrode was placed at 100 μm below the soma of the recorded neuron (at the level of layer 6), close to the basal dendrites of the recorded IL-L5PNs. Paired pulses (100 μs in duration and 20–100 μA in intensity, 50 ms away) were continuously delivered at 0.33 Hz. EPSCs were isolated in the presence of $GABA_AR$ (50 μM picrotoxin; PiTX) and $GABA_BR$ (5 μM CGP-55845) antagonists. IPSCs were isolated adding AMPAR (20 μM CNQX) and NMDAR (50 μM D-AP5) antagonists. In both cases, after 5 min of stable baseline, we superfused IGF-1 for 35 min to check for long-term synaptic plasticity by analyzing the EPSCs and IPSCs peak amplitudes. In current-clamp mode, after a stable baseline of PSPs, the stimulation intensity was increased until suprathreshold responses reached were ≈21% for 5 min. Next, we applied IGF-1 for 15 min and measured the number of APs; afterward, we returned to initial stimulation intensity and measured the amplitude of PSPs to study the effect of IGF-1 on synaptic transmission. Miniature EPSCs (mEPSCs) were recorded at −60 mV in ACSF in the presence of 1 μM TTX, 20 μM PiTX, and 5 μM CGP-55845 to isolate excitatory synaptic transmission.

## Conditioned fear

Rats were anesthetized with ketamine (70 mg/kg i.p.; Ketolar), xylazine (5 mg/kg i.p.; Rompum), and atropine (0,05 mg/kg i.p.; B. Braun Medical S.A) and maintained with isoflurane (2–3% in oxygen).

Animals were positioned in a stereotaxic apparatus (David Kopf Instruments, Tujunga, CA, USA) and placed on a water-heated pad at 37°C. The midline of the scalp was sectioned and retracted, and small holes were drilled in the skull. Rats were implanted with a single 26 gauge stainless-steel guide cannula (Plastics One) in the mPFC. Stereotaxic coordinates aiming toward the IL were (AP: 2.8 mm, LM: 1.0 mm, and DV: 4.1 mm) from bregma according to rat brain atlas (*Paxinos and Watson, 2007*), with the cannula angled 11° toward the midline in the coronal plane as described previously (*Santini et al., 2004*). A 33-gauge dummy cannula was inserted into the guide cannula to prevent clogging. Guide cannulas were cemented to the skull with dental acrylic (Grip Cement) and the incision was sutured. Buprenorphine hydrochloride (75 mg/kg s.c.; Buprex) was administered for post-surgical analgesia. Rats were allowed at least 7 days for surgery recovery. Trace fear conditioning was conducted in a chamber of 25 × 31 × 25 cm with aluminum and Plexiglas walls (Coulbourn, Allentown, PA). The floor consisted of stainless-steel bars (26 parallel steel rods 5 mm diameter, 6 mm spacing) that can be electrified to deliver a mild shock. A speaker was mounted on the outside wall, and illumination was provided by a single overhead light (miniature incandescent white lamp 28 V). The rectangular chamber was situated inside a sound-attenuating box (Med Associates, Burlington, VT) with a ventilating fan, which produced an ambient noise level of 58 dB. The CS was a 4 kHz tone with a duration of 30 s and an intensity of 80 dB. The US was a 0.4 mA scrambled footshock, 0.5 s in duration, which co-terminated with the tone during the conditioning phase. Between sessions, floor trays and shock bars were cleaned with soapy water and the chamber walls were wiped with a damp cloth. An additional Plexiglas chamber served as a novel context for the auditory cue test. This chamber, which was a triangle with a black smooth Plexiglas floor, was physically distinct from the fear conditioning chamber. All the environmental conditions were completely different regarding the fear conditioning test except for the ventilating fan. Before placing rats into this chamber, the chamber floor and walls were wiped with 30% vanilla solution to provide a background odor distinctive from that used during fear conditioning. The activity of each rat was recorded with a digital video camera mounted on top of each behavioral chamber. 30 min before extinction training, saline (NaCl 0.9%), IGF-1 (10 μM; Preprotech), or the IGF-1R antagonist 7-[*cis*-3-(1-azetidinylmethyl) cyclobutyl]−5-[3-(phenylmethoxy)phenyl]−7H-pyrrolo[2,3-]pyrimidin-4-amine (NVP-AEW 541, 40 μM; Cayman Chemicals) plus IGF-1 were infused into the IL. For the infusions, cannula dummies were removed from guide cannulas and replaced with 33-gauge injectors, which were connected by polyethylene tubing (PE-20; Small Parts) to 100 μl syringes mounted in an infusion pump (Harvard Apparatus). Drugs were infused at a rate of 0.5 μl/min for 1 min as described previously (*Fontanez-Nuin et al., 2011*).

## Non-cannulated animals

Animals were divided into two groups, the conditioned group (COND) and the extinguished group (EXT). On day 1, rats received three habituation trials (tone-no shock; habituation phase) into two different contexts (context 1: square shock box and context 2: triangle box with soft floor). On day 2, rats received three conditioning trials (tone paired with shock; context 1; condition phase). On day 3, rats of COND group remained in their home cage, whereas rats of EXT group received 20 tone-alone trials (context 2; extinction phase). On day 4, both groups of rats received five tone-alone trials to test for recall of conditioning or extinction (test phase) (*Figure 4—figure supplement 1*).

## Cannulated animals

Animals were divided into the saline group (SAL) and the different drug groups (IGF-1 or NVP + IGF-1). On day 1, rats received three habituation trials into the two contexts aforementioned. On day 2, rats received three conditioning trials. On day 3, rats were infused with saline or drug 30 min before the beginning of the extinction phase. On day 4, all groups of rats received five tone-alone trials in the same chamber (context 2) to test for recall of extinction (*Figure 4B*). In all phases of the experiment, the interval between successive tones was variable, with an average of 2 min. All groups were tested on the same day to determine the long-term changes occurring in the mPFC that gate subsequent memory retrieval.

## Data analysis

Data were analyzed using pClamp (Molecular Devices), Excel (Microsoft), and GraphPad Prism 8.3 software. Twenty responses were averaged except when otherwise indicated. The magnitude of the change in peak EPSCs, IPSCs, and PSPs amplitude was expressed as a percentage (%) of the baseline control amplitude and plotted as a function of time. The presynaptic or postsynaptic origin of the observed regulation of EPSCs and IPSCs amplitudes was tested by estimating the PPR changes, which were considered to be of presynaptic origin (*Clark et al., 1994*; *Creager et al., 1980*; *Kuhnt and Voronin, 1994*) and were quantified by calculating a PPR index (R2/R1), where R1 and R2 were the peak amplitudes of the first and second synaptic currents, respectively. To estimate the modifications in the synaptic current variance, we first calculated the noise-free coefficient of variation (CVNF) for the synaptic responses before and 40 min after applying IGF-1 in the bath with the formula $CVNF = \sqrt{(\delta XPSC^2 - \delta noise^2)}/m$; $\delta XPSC^2$ and $\delta noise^2$ (X = E excitatory or X = I inhibitory) are the variances of the peak EPSC or IPSC and baseline, respectively, and m is the mean EPSC or IPSC peak amplitude. The ratio of the CV measured before and 40 min after applying IGF-1 (CVr) was obtained for each neuron as CV after IGF-1 responses/CV control (*Clements, 1990*). Finally, we constructed plots comparing variation in M (m after IGF-1 responses over m at control conditions) against the changes in response variance of the EPSC or IPSC amplitude (1/CVr2) for each cell. In these plots, values were expected to follow the diagonal if the EPSC or IPSC depression had a presynaptic origin. Results are given as mean ± SEM (N = numbers of cells). There were no gender differences in our experiments. MiniAnalysis software (SynaptoSoft Inc) and pCLAMP software were used for the analysis of the frequency and amplitude of mEPSCs. See *Supplementary file 1* for a detailed description of the statistical analyses performed in these experiments.

The total freezing time during the 30 s tone was measured and converted to a percentage of freezing. Freezing was defined as the cessation of all movements except respiration. The percentage of freezing time was used as a measure of conditioned fear (*Blanchard and Blanchard, 1972*). The behavioral data were analyzed by manual evaluation of the videos and/or using the ImageJ free software with the plugging developed for this purpose (*Shoji et al., 2014*). No differences were observed with both methods. Values are reported as the means ± SEM. See *Supplementary file 1* for a detailed description of the statistical analyses performed in these experiments.

Blind experiments were not performed in the study but the same criteria were applied to all allocated groups for comparisons. Randomization was not employed. The sample size in whole-cell recording experiments was based on the values previously found sufficient to detect significant changes in past studies from the lab.

Statistical analysis was performed using Prism 9 (GraphPad). Prior to any statistical analysis, we assessed normality distribution of our data using Kolmogorov–Smirnov test for normality. Then, we compared data means using t-test or Wilcoxon test for comparisons of two groups. Multiple t-test or ANOVA (one-way or two-way) test for more than two groups for parametric distributions and Mann–Whitney for non-parametric distributions. Specific statistical tests and number of animals are indicated in each figure legend.

## Acknowledgements

This work was supported by the following Grants: BFU2013-43668-P and BFU2016-0802-P AEI/FEDER, UE (MINECO) to D Fernández de Sevilla. We thank Dr. Washington Buño, Ricardo Gomez and Ignacio Torres-Aleman for helpful comments and Dr. Cesar Venero for his advice and help on the conditioned fear experiments.

## Additional information

### Funding

| Funder | Grant reference number | Author |
| --- | --- | --- |
| Ministerio de Economía, Industria y Competitividad, Gobierno de España | BFU2013-43668-P | David Fernández de Sevilla |

| Ministerio de Economía, Industria y Competitividad, Gobierno de España | BFU2016-0802-P AEI/FEDER | David Fernández de Sevilla |
| Ministerio de Economía, Industria y Competitividad, Gobierno de España | UE | David Fernández de Sevilla |

The funders had no role in study design, data collection and interpretation, or the decision to submit the work for publication.

## Author contributions

Laura E Maglio, Conceptualization, Data curation, Formal analysis, Investigation, Methodology, Writing - original draft, Writing - review and editing; José A Noriega-Prieto, Conceptualization, Data curation, Formal analysis, Investigation, Methodology, Writing - review and editing; Irene B Maroto, Jesús Martin-Cortecero, Antonio Muñoz-Callejas, Formal analysis, Investigation; Marta Callejo-Móstoles, Investigation; David Fernández de Sevilla, Conceptualization, Formal analysis, Supervision, Funding acquisition, Validation, Investigation, Visualization, Methodology, Writing - original draft, Project administration, Writing - review and editing

## Author ORCIDs

Laura E Maglio (iD) https://orcid.org/0000-0003-4760-5624
José A Noriega-Prieto (iD) https://orcid.org/0000-0002-3086-8746
Irene B Maroto (iD) https://orcid.org/0000-0002-4156-166X
Jesús Martin-Cortecero (iD) http://orcid.org/0000-0002-9350-6267
David Fernández de Sevilla (iD) https://orcid.org/0000-0001-6344-0773

## Ethics

Animal experimentation: All animal procedures were approved by the Universidad Autónoma of Madrid Ethical Committee on Animal Welfare and conform to Spanish and European guidelines for the protection of experimental animals (Directive 2010/63/EU). An effort was made to minimize animal suffering and number.

## Decision letter and Author response

Decision letter https://doi.org/10.7554/eLife.67267.sa1
Author response https://doi.org/10.7554/eLife.67267.sa2

## Additional files

### Supplementary files

• Supplementary file 1. Summary of statistical significance and tests used in each figure.

• Transparent reporting form

### Data availability

All data generated or analysed during this study are included in the manuscript and supporting files.

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
