## [Decision Letter]

**Acceptance summary:**

This paper shows that insulin growth factor-1 (IGF-1) plays an important role in regulating intrinsic excitability and synaptic transmission in the infralimbic cortex, facilitating the extinction of fear memory.

**Decision letter after peer review:**

[Editors’ note: the authors submitted for reconsideration following the decision after peer review. What follows is the decision letter after the first round of review.]

Thank you for submitting your work entitled "IGF-1 facilitates extinction of conditioned fear" for consideration by *eLife*. Your article has been reviewed by 3 peer reviewers, one of whom is a member of our Board of Reviewing Editors, and the evaluation has been overseen by a Reviewing Editor and a Senior Editor. The reviewers have opted to remain anonymous.

Our decision has been reached after consultation between the reviewers. Based on these discussions and the individual reviews below, we regret to inform you that your work will not be considered further for publication in *eLife*.

While reviewers think that the findings linking behavioral and neuronal changes by IGF1 are of potential interest, they pointed to numerous methodological, statistical and biological problems (see the detailed comments below).

Reviewer #1:

This work comes to show the role IGF1 plays in memory extinction. The authors suggest that IGF1 increases intrinsic excitability of L5P neurons in the infralimbic cortex (IL) by decreasing AHP. In addition, it attenuates both IPSCs and EPSCs by presynaptic mechanisms and facilitates a postsynaptic potentiation. Finally, the authors show that animals injected with IGF1 intracranially exhibit increased extinction of fear memory, and that IL-L5P neurons of these animals have reduced AHP current, increased excitability and decreased mEPSC frequency. The authors concluded that IGF1 regulates the cellular and synaptic processes underling fear memory extinction in the IL cortex.

1. Throughout the work the authors show that application of IGF1 causes the aforementioned effects, while they did not address the question whether IGF1 has a physiological role in excitability, synaptic transmission and behavioral phenotypes. It is unclear whether inhibition of IGF1R by NVP-AEW541 caused a decrease in mAHP and sAHP. Moreover, from Fig. S1 it looks like that the IGF1R antagonist has an inhibitory effect on fAHP. The authors should analyze the effect of IGF1R antagonist and show statistics for both, electrophysiological and behavioral phenotypes.

2. The authors show that IGF1 application causes a long-term depression of sAHP, while did not identify the mechanism mediating it. It would be important to identify the channel involved in AHP regulation. Moreover, it would be important to show that block of this channel occludes an increase in I-F ratio. Otherwise, the authors should acknowledge that AHP changes and I-F changes could be mediated by distinct mechanisms.

3. The authors show that IGF1 increases PSP (Fig. 3I). They increased the intensity of stimulation to get some supra-threshold events in the middle of the protocol. The authors must show that this protocol does not cause a potentiation of PSP by itself, without addition of IGF1.

4. Previous study found that extinction causes a decrease in fAHP, but not in sAHP in the IL (Santini et al., 2008). In contrast, conditioning up-regulated sAHP in that study. Moreover, a decrease in fAHP by M-channel inhibitor facilitated extinction (Santini and Porter, 2010). How do the authors explain this discrepancy between the results?

Reviewer #2:

This paper examines the effects of insulin-like growth factor-1 (IGF-1) on intrinsic excitability and synaptic transmission in neurons from the infralimbic cortex and on the extinction of fear memory. The authors report an increase in excitability mediated by the reduction of mAHP and a reduction in excitatory and inhibitory synaptic transmission. They also show that IGF-1 facilitates the extinction of fear memory.

This paper is interesting however, there are many issues. First, IGF-1 application is rather artificial and therefore this does not correspond to a natural way of stimulating neurons. In addition, some protocols and measures must be clarified.

It is not clear what the authors precisely measured when the declare measuring mAHP and sAHP. In contrast with what seems inferred by the authors the medium and slow AHPs can be measured with the same depolarization. In addition, the sIAHP generally peaks at ~0.5-1.0 s while their records display only a current peaking at ~100 ms. So there is no sAHP in the data provided in Figure 1 and 2 and therefore, conclusion about the effect of IGF-1 on sAHP is erroneous.

Effects of IGF-1 on PSCs. The application of IGF-1 is very long (>30 min) and the test time after application is too short (~15 minutes) to really consider the depression as long-lasting. The authors should consider shorter application times (5 or 10 min) and longer test times (at least 30 min) to know whether the persistent effect is due or not to an incomplete elimination of IGF-1 applied in the bath.

Does the effect of IGF-1 on intrinsic excitability long-lasting? To test this, brief application of IGF-1 should be used too.

The rational of the experiment depicted in Fig. 3I en 3J is not clear at all. Please provide details why stimulus intensity is increased during 5 min to obtain 21% of supratheshold responses.

Reviewer #3:

In this manuscript, the authors explore the involvement of IGF-1 in learning, focusing on fear extinction. They show that IGF-1 modulates network and intrinsic firing properties of L5, increasing firing frequency via inhibition of AHP and affecting network properties by reducing the release probability of presynaptic neurons. In addition, they demonstrate a concomitant increase of the firing probability in the postsynaptic L5 neurons. Intriguingly these effects develop rather slowly, with a plateau reached after ~40 min. They further show that in vivo administration of IGF-1 during the extinction phase results in enhanced fear extinction, correlating with the long-lasting effect, of over 24 h, of AHP inhibition and increased firing.

Overall these findings are interesting, linking behavioral and neuronal changes. Nevertheless, important experimental data, that were collected during the performed experiments, are not presented, leaving the reader with a partial view (see below in the comments). Moreover, some controls are missing (see below) and the number of cells in the electrophysiological recordings is rather low (n = 4/5/6 in many key figures).

1. Apart from changes in AHP, were there any additional changes in single AP properties? Did you observe any changes in AP threshold? Kinetics of upstroke or decay? Max amp? Duration? Input resistance?

The examples shown in Fig 1E depicts similar AHP with repetitive firing. If so, additional alteration in AP properties may be responsible for increased firing following IGF-1 application.

2. Fig. 1 - AHP amplitudes are presented as deltas, but the reference point is unclear, was it AP threshold? Also, since these are paired recordings, an illustration highlighting the pairs can be useful.

Were there differences in the time to max AHP? Were there differences between the % change in mAHP and sAHP for each cell? A similar % reduction in both may indicate a common mechanism, while the lack of correlation between mAHP and sAHP may hint that multiple mechanisms are at play. How long did you perfuse IGF-1? (~40 min?) what was the kinetics of this effect?

3. Fig 1B, D. The variability seems to be lower in the NVP group compare to the ACSF group. Could that be attributed to the variable levels of endogenous IGF-1?

4. Fig 1E. Did you observe AHP alterations during prolonged firing? the presented examples depict similar AHP levels after each AP and the potential at the end of the pulse is slightly more hyperpolarized after IGF1 addition, in contrast to Fig. 1A. Are there any other parameters that might support enhanced firing?

5. How many rats were used for each data set? this is not indicated throughout the manuscript.

6. The slow kinetics of the AHP and synaptic currents response to IGF1, as well as the long-lasting effect observed following IGF-1 washout (in slices and in vivo), should be discussed. Such slow kinetics may suggest the possible involvement of protein synthesis or insertion/endocytosis.

7. Line 273 - did you mean "did not require depolarization"? Sup Fig 2 depicts sAHP reduction of over 60% without depolarization. The time courses, with or without stimulation, are somewhat different, but the endpoints (after 50 min) are similar (showing maybe even a bit more inhibition without depolarization). Thus, depolarization may facilitate the process, but is not a prerequisite.

8. Lines 288-290 - "Additionally, this modulation was not observed when the synaptic stimulation was absent (Figure 3D), suggesting that the evoked synaptic responses were required for this LTD of the EPSCs". This statement is confusing and is not supported by the data. Comparing Fig. 3A and 3D, the endpoints (after 50 min) are similar, with inhibition levels of 30-40%. Thus, synaptic stimulation during IGF-1 application is not required for presynaptic LTD. Despite that, as for the kinetics of AHP, depolarization may in fact facilitate this effect. Thus, two mechanisms may coexist: one that supports EPSC inhibition with synaptic activation, with faster kinetics that plateaus within 10 min of IGF1 application, and another slower process, independent of synaptic activation, which plateaus at 50 min. This point requires further clarification and discussion.

9. For IPSCs, the kinetics is similar with or without stimulation. But, the level of inhibition after 50 min is doubled without synaptic stimulation. These properties should also be further discussed.

10. Fig 3I-J - These results are very interesting and important, as mimicking of firing by synaptic activation takes into account changes in multiple presynaptic neurons (excitatory and feedforward inhibition), as well as postsynaptic changes. Indeed, firing in response to current injection into the soma is less physiological. Moreover, SK channels in different cellular locations were shown to confer contradicting effects on EPSPs amplitude and duration, and thus on neuronal excitability (studied by the Stuart lab as well as others). However, valuable information is missing here as well. Did you see changes in PSP duration? changes in feedforward inhibition? AP properties? do you see similar AHP inhibition also with synaptic activation?

The opposing effects observed in voltage clamp and current clamp modes (reduced EPSP amplitude with increased PSP depolarization) need to be discussed. While one possibility for this can be a change in the input resistance, another may reside in alterations of the interplay between dendritic SK and NMDA channels (see Bock et al. 2019). Also, the mechanism for enhanced firing with synaptic stimulation may be different from increased firing with current injection. In addition to the site of depolarization, synaptic stimulations were given at low freq , as oppose the high frequency firing with prolonged depolarization. While the rate of synaptic stimulation is not specified for the current clamp experiments shown (should be added please), the graphical representations suggest 1 stim/min. What was the duration of IGF-1 application here? The methods section indicates 15 min, but the graph and legend specify 35 min. The kinetics for PSP potentiation seems to be faster, compared to the VC experiments. Any indication of the underlying cause?

11. Overall the method section is missing some vital information. Was the age of the rats used for behavioral experiments also P20 - P30? were the surgeries performed at P20? with 7 days recovery the behavioral experiments supposedly spanned P28 - 31, correct? what were the intervals between each step of the trials (cond, ext, test)?

12. Fig 4 - What was the effect of NVP alone? this control is vital. IGF-1R was shown to be fully activated at baseline conditions (Gazit et al. 2016), thus, with NVP and IGF-1, some R might still be functional and activated by IGF1, resulting in partial inhibition by NVP. Moreover, the main conclusion here seems to be drawn from the relatively low levels of fear extinction in the saline group (note the typo - extintion). Comparing the data in Fig 4 to Sup Fig 3, sup Fig 3 illustrates what seems to be further extinction during the extinction phase, as well as during the test phase. Reduction in freezing is missing in the saline group shown in Fig 4. What might reconcile these observations?

13. Was there a correlation between the sAHP amplitude and freezing behavior? One might expect lower sAHP amp in rats with better extinction levels.

14. Did you also observe lower levels of mAHP after in vivo application of IGF1/NVP? any effect on fAHP? is it possible to see a further reduction of sAHP with acute bath application of IGF-1 in mice that were pre-treated with IGF-1 during fear extinction? this would reveal the dynamic range of this long-lasting effect.

15. The variability in mEPSC is large, especially in the IGF-1 group. Recordings from more cells can fix that.

16. The molecular mechanism is not sufficiently addressed. The authors suggest that SK-, m-currents or mGluR5 might be involved. However, no direct evidence for the specific involvement of these channels is provided, nor any explanation for the slow and long-lasting effect observed.

[Editors’ note: further revisions were suggested prior to acceptance, as described below.]

Thank you for resubmitting your work entitled "IGF-1 facilitates extinction of conditioned fear" for further consideration by *eLife*. Your revised article has been evaluated by John Huguenard (Senior Editor) and a Reviewing Editor.

The manuscript has been improved but there are some remaining issues that need to be addressed, as outlined below:

1. The authors performed new experiments showing that inhibition of IGF1Rs occludes the effects of exogenously applied IGF1, but does not cause any effect on neuronal/synaptic function and memory extinction. This raises the question whether endogenous IGF1 plays a physiological role in these processes? In the discussion the authors claim that "....IGF-1 appears as a key endogenous molecule in the modulation of the extinction of conditioned fear memory, supporting the role of IGF-1 as a crucial piece in behavioral tasks." Unfortunately, the results do not support this central claim.

We would like to ask the authors to modify these claims and to provide some additional discussion on the relevance of the results to some conditions with elevated cerebral IGF1 levels.

2. The authors performed new experiments showing that inhibition of G protein activation by GDPβs prevents the reduction in mAHP, while had no effect on sAHP (Fig. 2F-G). First, in the Fig. 2G the mAHP seems to be altered as well. The p-value for this comparison is 0.06, and an additional single cell may render GDPβs ineffective at all at preventing the effect of IGF1 on AHP (6 cells per group seems rather low). Second, the authors wrote in their discussion: "In our study, inhibition of G protein-coupled receptors by GDPβs abolished the IGF-1-mediated reduction of mIAHP, suggesting that the effect of IGF-1 on mAHP could be mediated by the activation of mGluR5." This conclusion is far reaching since there are many other possible mechanisms may explain this effect. The authors should tone down this claim since no evidence on mGluR5 involvement is provided.

3. Figure 5, which is supposed to present mEPSC data, is missing.

---

## [Author Response]

[Editors’ note: the authors resubmitted a revised version of the paper for consideration. What follows is the authors’ response to the first round of review.]

Reviewer #1:This work comes to show the role IGF1 plays in memory extinction. The authors suggest that IGF1 increases intrinsic excitability of L5P neurons in the infralimbic cortex (IL) by decreasing AHP. In addition, it attenuates both IPSCs and EPSCs by presynaptic mechanisms and facilitates a postsynaptic potentiation. Finally, the authors show that animals injected with IGF1 intracranially exhibit increased extinction of fear memory, and that IL-L5P neurons of these animals have reduced AHP current, increased excitability and decreased mEPSC frequency. The authors concluded that IGF1 regulates the cellular and synaptic processes underling fear memory extinction in the IL cortex.1. Throughout the work the authors show that application of IGF1 causes the aforementioned effects, while they did not address the question whether IGF1 has a physiological role in excitability, synaptic transmission and behavioral phenotypes. It is unclear whether inhibition of IGF1R by NVP-AEW541 caused a decrease in mAHP and sAHP. Moreover, from Fig. S1 it looks like that the IGF1R antagonist has an inhibitory effect on fAHP. The authors should analyze the effect of IGF1R antagonist and show statistics for both, electrophysiological and behavioral phenotypes.

We are grateful to the reviewer for these perceptive comments. We have performed additional statistical analyses to address the reviewer´s concerns. We analysed the effect of IGF-1R antagonist and added the statistical values in the figures, figure legends and supplement table 1, for all experiments performed with IGF-1R antagonist (Figure 1, Figure 2, Figure Supplementary 1, and Figure 4). We did not observe significant differences between ACSF (control condition) and NVP, which discards an inhibitory effect on AHP (fast, medium and slow) by the antagonist of IGF1R (Supplementary Table 1).

2. The authors show that IGF1 application causes a long-term depression of sAHP, while did not identify the mechanism mediating it. It would be important to identify the channel involved in AHP regulation. Moreover, it would be important to show that block of this channel occludes an increase in I-F ratio. Otherwise, the authors should acknowledge that AHP changes and I-F changes could be mediated by distinct mechanisms.

As the reviewer suggested, we did experiments analysing the involvement of metabotropic receptors by blocking G protein activation. As shown in Figure 2F-G, preventing G protein activation by GDPbs in the patch pipette blockade of metabotropic receptor signaling and prevented the reduction of mAHP by IGF-1 while it had no effect on sAHP. These results indicate that the reduction of both AHPs currents would be mediated by different mechanisms and that metabotropic glutamate receptors could be responsible for the reduction of mAHP as previously demonstrated by Young et al. 2007 in the hippocampus.

3. The authors show that IGF1 increases PSP (Fig. 3I). They increased the intensity of stimulation to get some supra-threshold events in the middle of the protocol. The authors must show that this protocol does not cause a potentiation of PSP by itself, without addition of IGF1.

We agree and we show that the protocol does not cause any change in the PSP by itself, without addition of IGF-1 ( see new Figure supplementary 4).

4. Previous study found that extinction causes a decrease in fAHP, but not in sAHP in the IL (Santini et al., 2008). In contrast, conditioning up-regulated sAHP in that study. Moreover, a decrease in fAHP by M-channel inhibitor facilitated extinction (Santini and Porter, 2010). How do the authors explain this discrepancy between the results?

We have performed additional analysis to address the reviewer´s concerns.

First, we have performed analysed of fAHP from NAÏVE rats and all groups of extinction rats (EXT, Ext-Saline, Ext-IGF-1 and Ext-NVP+IGF-1) (new Figure supplementary 6A). The results show a reduction on fAHP in all animals that were exposed at extinction protocols, what is in agreement with previous study published by Santini et al. 2008. We could not include the conditioned group in the statistical analysis since we did not perform electrophysiology experiment in this group. Our object of study was the effect of IGF-1 on the fear extinction memory. Following the reviewer´s comment, we also analysed sAHP from all groups (new Figure supplementary 6C) and we find no significant differences among NAÏVE rats and all extinctions groups, except to IGF-1-treat group. Thus these results do not contradict the previous study published by Santini et. al. 2008. The significant reduction on sAHP from IGF-1-treat group compared with NAÏVE indicates an additional effect of IGF-1 on the extinction.

Reviewer #2:This paper examines the effects of insulin-like growth factor-1 (IGF-1) on intrinsic excitability and synaptic transmission in neurons from the infralimbic cortex and on the extinction of fear memory. The authors report an increase in excitability mediated by the reduction of mAHP and a reduction in excitatory and inhibitory synaptic transmission. They also show that IGF-1 facilitates the extinction of fear memory.This paper is interesting however, there are many issues. First, IGF-1 application is rather artificial and therefore this does not correspond to a natural way of stimulating neurons. In addition, some protocols and measures must be clarified.

Although we use a long lasting application of IGF-1 in the manuscript we have used shorter application showing similar results ( see new Figure supplementary 5). Moreover, in the CNS, it has been described that neuronal activity produces both an increase in IGF-1 release (Cao et al., 2011) and the entry of serum IGF-1 into the CNS through the blood– brain barrier (Nishijima et al., 2010). Therefore, during some physiological conditions in which serum IGF-1 levels are increased after physical exercises, a long-lasting increase in the brain areas showing neuronal AP activity is expected. The long lasting IGF-1 application in our manuscript was trying to mimic this natural condition. We have performed new analyses and the description of how the measurements were done has been explained in detail in the new version of the Materials and Methods section.

It is not clear what the authors precisely measured when the declare measuring mAHP and sAHP. In contrast with what seems inferred by the authors the medium and slow AHPs can be measured with the same depolarization. In addition, the sIAHP generally peaks at ~0.5-1.0 s while their records display only a current peaking at ~100 ms. So there is no sAHP in the data provided in Figure 1 and 2 and therefore, conclusion about the effect of IGF-1 on sAHP is erroneous.

We appreciate this constructive comment of the reviewer. We have included new analyses from electrophysiological experiments using the work of Santini and coworkers (Santini et al., 2008) as a reference. We have measured mIAHP as the current peak and sIAHP as the area under the curve for a 1-s period after the current peak. We add a new version of Figure 1, Figure 2, Figure Supplementary 3, and Figure 4. Also, we added a description of how these measurements were done in the Materials and Methods section. Our results show that IGF-1 leads to a reduction in the mAHP and sAHP in IL-L5PN in an IGF-1R activation-dependent manner.

Effects of IGF-1 on PSCs. The application of IGF-1 is very long (>30 min) and the test time after application is too short (~15 minutes) to really consider the depression as long-lasting. The authors should consider shorter application times (5 or 10 min) and longer test times (at least 30 min) to know whether the persistent effect is due or not to an incomplete elimination of IGF-1 applied in the bath.

We agree with the reviewer and we have performed new experiments in which IGF-1 was applied during 10 minutes (see new Figure supplementary 5). The results now clearly shows that IGF-1 induced a long-lasting potentiation of the synaptic transmission that persists after the complete elimination of IGF-1 in the bath .

Does the effect of IGF-1 on intrinsic excitability long-lasting? To test this, brief application of IGF-1 should be used too.

We performed new experiments and evaluated the intrinsic excitability of L5PNs of the IL after 10 min of IGF-1 application. We observed a slight increase in the excitability that is maintained at 60 min although it did not reach statistical significance. Therefore, we decided not to speculate about this possibility in the manuscript.

The rational of the experiment depicted in Fig. 3I en 3J is not clear at all. Please provide details why stimulus intensity is increased during 5 min to obtain 21% of supratheshold responses.

As we state in our response to the major point 1 of this reviewer, we were trying to mimic a possible situation in which AP based activity in L5PNs would induce the increase of the levels of IGF-1 and by entry of serum IGF-1 into the IL through the blood–brain barrier. Therefore, the rational of the experiment was to set an AP activity that did not induce plasticity per se but that could be used to induce it after the increase in the IGF-1 levels, trying to mimic the AP dependent entry of serum IGF-1 into a brain region due to its activity occurring in natural conditions.

Reviewer #3:In this manuscript, the authors explore the involvement of IGF-1 in learning, focusing on fear extinction. They show that IGF-1 modulates network and intrinsic firing properties of L5, increasing firing frequency via inhibition of AHP and affecting network properties by reducing the release probability of presynaptic neurons. In addition, they demonstrate a concomitant increase of the firing probability in the postsynaptic L5 neurons. Intriguingly these effects develop rather slowly, with a plateau reached after ~40 min. They further show that in vivo administration of IGF-1 during the extinction phase results in enhanced fear extinction, correlating with the long-lasting effect, of over 24 h, of AHP inhibition and increased firing.Overall these findings are interesting, linking behavioral and neuronal changes. Nevertheless, important experimental data, that were collected during the performed experiments, are not presented, leaving the reader with a partial view (see below in the comments). Moreover, some controls are missing (see below) and the number of cells in the electrophysiological recordings is rather low (n = 4/5/6 in many key figures).1. Apart from changes in AHP, were there any additional changes in single AP properties? Did you observe any changes in AP threshold? Kinetics of upstroke or decay? Max amp? Duration? Input resistance?The examples shown in Fig 1E depicts similar AHP with repetitive firing. If so, additional alteration in AP properties may be responsible for increased firing following IGF-1 application.

We have now analyzed the AP characteristics and there are no differences in all the parameters studied (new Figure supplementary 2A-G). However, the application of IGF-1 increased the membrane resistance (Rm) (see new Figure supplementary 2H), suggesting that IGF-1 reduces ionic conductances. This result is in agreement with the IGF-1-mediated reduction of mAHP and sAHP currents.

2. Fig. 1 - AHP amplitudes are presented as deltas, but the reference point is unclear, was it AP threshold? Also, since these are paired recordings, an illustration highlighting the pairs can be useful.

AHP amplitudes were measured as the membrane potential variation from the resting membrane potential. This has now been explained in the Materials and Methods section and the legend of Figure 1. There were not paired recordings. The recordings were performed before and after the IGF-1 application.

3. Were there differences in the time to max AHP? Were there differences between the % change in mAHP and sAHP for each cell? A similar % reduction in both may indicate a common mechanism, while the lack of correlation between mAHP and sAHP may hint that multiple mechanisms are at play. How long did you perfuse IGF-1? (~40 min?) what was the kinetics of this effect?

We thank the Reviewer for this insightful suggestion. We reanalyzed the data and measured the mAHP current at the peak of the current and the sAHP current as of the area under the curve during 1 s period beginning at the peak of the current after a depolarizing pulse of 800 ms, and these values were included in the new version of the manuscript. We also compared the percentage of change for mIAHP and sIAHP and there was no significant difference (32.8% vs. 39.5 %). IGF-1 was present in the bath for 40 min to measure its effect on the AHPs. The results reveal a slow kinetics reaching a stable reduction of AHP at 30 minutes (see Figure 2C,D and Figure supplementary 3 B-C).

4. Fig 1B, D. The variability seems to be lower in the NVP group compare to the ACSF group. Could that be attributed to the variable levels of endogenous IGF-1?

We have performed additional statistical analyses comparing ACSF vs NVP groups and we did not observe differences between them. This analysis has been included in the new version of the manuscript (see Supplementary Table 1).

5. Fig 1E. Did you observe AHP alterations during prolonged firing? the presented examples depict similar AHP levels after each AP and the potential at the end of the pulse is slightly more hyperpolarized after IGF1 addition, in contrast to Fig. 1A. Are there any other parameters that might support enhanced firing?

We have included a new representative example in figure 1G. As aforementioned (Essential revisions point 1), we observed no changes in AP characteristics, suggesting that the enhanced firing may be a consequence of the reduction in AHP currents.

6. How many rats were used for each data set? this is not indicated throughout the manuscript.

The number of rats used in each group of experiments has been included in figure legends. Now, the number of experiments is stated as the “number of cells/number of animals”.

7. The slow kinetics of the AHP and synaptic currents response to IGF1, as well as the long-lasting effect observed following IGF-1 washout (in slices and in vivo), should be discussed. Such slow kinetics may suggest the possible involvement of protein synthesis or insertion/endocytosis.

We now discuss this point in the new version of the manuscript (page 25): “Cohen-Matslish and coworkers showed that the AHP reduction induced by the synaptic activation requires protein synthesis during a specific time-window for its long-term maintenance (Cohen-Matsliah et al., 2010). It has been shown that the acquisition of extinction memory depends on protein synthesis (Santini et al., 2004). In our study, we show that IGF-1 favors the acquisition of extinction memory when applied directly to the IL in vivo. Therefore, it is tempting to hypothesize that the slow onset kinetics of AHP reduction that we observed after bath perfusion of IGF-1 in brain slices are suggestive of a mechanism involving protein synthesis, although further experiments would be needed to confirm it. Consistent with our observations in brain slices (plateau of AHP reduction is reached after 30 min), AHP amplitude is reduced in IL neurons from rats where the extinction memory was consolidated (recordings from the test day in IGF-1treated animals). Moreover, we found that a reduction in AHP amplitude was directly correlated with a reduction in the expression of fear, suggesting that mechanisms favoring a greater AHP reduction would produce an enhancement of the acquisition of extinction memory”.

8. Line 273 - did you mean "did not require depolarization"? Sup Fig 2 depicts sAHP reduction of over 60% without depolarization. The time courses, with or without stimulation, are somewhat different, but the endpoints (after 50 min) are similar (showing maybe even a bit more inhibition without depolarization). Thus, depolarization may facilitate the process, but is not a prerequisite.

We have corrected this issue in the new version of Figure supplementary 2, now Figure supplementary 3. We represent in the same graph the time course of the AHPs with a and without evoking the AHP during the application of IGF-1 during the first 20 min. We show that the % reduction achieved at 60 min is not statistically different. However, the time course is altered and there is a shift to the right indicating that depolarization is not a prerequisite but facilitates the modulation.

9. Lines 288-290 - "Additionally, this modulation was not observed when the synaptic stimulation was absent (Figure 3D), suggesting that the evoked synaptic responses were required for this LTD of the EPSCs". This statement is confusing and is not supported by the data. Comparing Fig. 3A and 3D, the endpoints (after 50 min) are similar, with inhibition levels of 30-40%. Thus, synaptic stimulation during IGF-1 application is not required for presynaptic LTD. Despite that, as for the kinetics of AHP, depolarization may in fact facilitate this effect. Thus, two mechanisms may coexist: one that supports EPSC inhibition with synaptic activation, with faster kinetics that plateaus within 10 min of IGF1 application, and another slower process, independent of synaptic activation, which plateaus at 50 min. This point requires further clarification and discussion.10. For IPSCs, the kinetics is similar with or without stimulation. But, the level of inhibition after 50 min is doubled without synaptic stimulation. These properties should also be further discussed.

After performing a new analysis, we agree with the reviewer on the coexistence of two mechanisms for the LTD of both the EPSC and IPSC. This possibility is now discussed in the new version of the manuscript accordingly (page 23): “Interestingly, we observed that two mechanisms coexist in the development of the LTD of EPSCs and IPSCs following IGF-1 application. The first one is dependent on synaptic activation and is characterized by a fast onset, the plateau being achieved at 10-15 min after IGF-1 application. The second one is independent of synaptic activation and is characterized by a slower onset, achieving the plateau at 40 min after IGF-1 application”.

Moreover we included the statistical analysis comparing the inhibition levels in the Figure 3A and 3D for EPSC and Figure 3E and 3H for IPSC (see Supplementary Table 1).

11. Fig 3I-J - These results are very interesting and important, as mimicking of firing by synaptic activation takes into account changes in multiple presynaptic neurons (excitatory and feedforward inhibition), as well as postsynaptic changes. Indeed, firing in response to current injection into the soma is less physiological. Moreover, SK channels in different cellular locations were shown to confer contradicting effects on EPSPs amplitude and duration, and thus on neuronal excitability (studied by the Stuart lab as well as others). However, valuable information is missing here as well. Did you see changes in PSP duration? changes in feedforward inhibition? AP properties? do you see similar AHP inhibition also with synaptic activation?

We have shown the lack of effect of IGF-1 in the AP properties and PSP duration (see Figure 3). However we did not study the modulation of the AHP evoked by synaptic activation nor the possible changes in the feedforward inhibition.

12. The opposing effects observed in voltage clamp and current clamp modes (reduced EPSP amplitude with increased PSP depolarization) need to be discussed. While one possibility for this can be a change in the input resistance, another may reside in alterations of the interplay between dendritic SK and NMDA channels (see Bock et al. 2019).

We have now discussed the possible mechanisms for the PSP potentiation in the manuscript (page 23): “However, the increase in the input resistance induced by IGF-1 could also explain the resultant LTP of the PSPs nonetheless of the reduction in the release of neurotransmitters. Since SK channels have been shown to modulate NMDA receptors (Jones et al., 2017; Ngo-Anh et al., 2005), we cannot discard a putative IGF-1dependent modulation of SK channels as the source of the potentiation in the synaptic transmission. The resultant LTP of the PSPs and the increase in the input resistance summed to the decrease of the AHPs would result in the increase of the L5PN activity of the IL and its effect on the fear extinction memory”.

13. Also, the mechanism for enhanced firing with synaptic stimulation may be different from increased firing with current injection. In addition to the site of depolarization, synaptic stimulations were given at low freq , as oppose the high frequency firing with prolonged depolarization. While the rate of synaptic stimulation is not specified for the current clamp experiments shown (should be added please), the graphical representations suggest 1 stim/min.

Recordings were done at 0.33 Hz (1 stim each 3-s) and the average response of 20 recordings (1 min) was represented (e.g., Figure 3I). It is now stated in the Material and Methods section.

14. What was the duration of IGF-1 application here? The methods section indicates 15 min, but the graph and legend specify 35 min. The kinetics for PSP potentiation seems to be faster, compared to the VC experiments. Any indication of the underlying cause?

Thank you for noticing this inconsistence. We have now clarified that the duration of IGF-1 application was 35 minutes (see page 8, lines 139 and 141) in both voltage-clamp (VC) and current-clamp (CC). The development over time of PSP potentiation seems to be faster in CC compared to VC, which could be explained because of the increase in the membrane resistance (see page 14, line 262, and Figure supplementary 2H).

15. Overall the method section is missing some vital information. Was the age of the rats used for behavioral experiments also P20 - P30? were the surgeries performed at P20? with 7 days recovery the behavioral experiments supposedly spanned P28 - 31, correct? what were the intervals between each step of the trials (cond, ext, test)?

Material and Methods section has been extensively rewritten in the new version of the manuscript. For behavioral experiments, rats underwent surgery at postnatal day 21 (after weaning) and were allowed to recover for 7-9 days. Therefore, behavioral tests were started at postnatal day 28-30 and finished at postnatal day 32-34. This same day (P32-P34), animals were sacrificed, and the electrophysiological recordings were done. The interval between each step of the trial was 24 h. All this information is now stated in the Material and Methods section.

16. Fig 4 - What was the effect of NVP alone? this control is vital. IGF-1R was shown to be fully activated at baseline conditions (Gazit et al. 2016), thus, with NVP and IGF-1, some R might still be functional and activated by IGF1, resulting in partial inhibition by NVP. Moreover, the main conclusion here seems to be drawn from the relatively low levels of fear extinction in the saline group (note the typo - extintion). Comparing the data in Fig 4 to Sup Fig 3, sup Fig 3 illustrates what seems to be further extinction during the extinction phase, as well as during the test phase. Reduction in freezing is missing in the saline group shown in Fig 4. What might reconcile these observations?

Although we agree with the reviewer that the application of IGF-1 and NVP could result in partial inhibition of IGF-1R by NVP, this partial inhibition is sufficient to demonstrate the involvement of IGF-1R activation in the modulation caused by IGF-1. We agree that the saline group shows a relatively low levels of fear extinction, however IGF-1 is able to increase this levels of fear extinction.

17. Was there a correlation between the sAHP amplitude and freezing behavior? One might expect lower sAHP amp in rats with better extinction levels.

We have analyzed the correlation between the mAHP/sAHP and the percentage of freezing (see new Figure 4F). The correlation parameters are R = 0.702 for mAHP and R = 0.787 sAHP. This correlation is now discussed in the manuscript (page 25): “Moreover, we found that a reduction in AHP amplitude was directly correlated with a reduction in the expression of fear, suggesting that mechanisms favoring a greater AHP reduction would produce an enhancement of the acquisition of extinction memory”.

18. Did you also observe lower levels of mAHP after in vivo application of IGF1/NVP? any effect on fAHP? is it possible to see a further reduction of sAHP with acute bath application of IGF-1 in mice that were pre-treated with IGF-1 during fear extinction? this would reveal the dynamic range of this long-lasting effect.

a. We did not observe lower levels of mAHP after in vivo application ofNVP+ IGF-1 but we did observe lower levels of mAHP in the in vivo IGF-1 group (see new Figure 4D).

b. There were no significant differences between in vivo fAHP groups where the extinction protocol was performed (see Figure supplementary 6).

c. Although a further reduction of sAHP with acute bath application ofIGF-1 in mice that were pre-treated with IGF-1 during fear extinction would add new information about the dynamic range of the long-lasting effect of IGF-1 in the modulation of the sAHPs, in our opinion this is out of the scope of the present manuscript.

19. The variability in mEPSC is large, especially in the IGF-1 group. Recordings from more cells can fix that.

Although we agree with the reviewer that the application of IGF-1 and NVP could result in partial inhibition of IGF-1R by NVP, this partial inhibition is sufficient to demonstrate the involvement of IGF-1R activation in the modulation caused by IGF-1. We agree that the saline group shows a relatively low levels of fear extinction, however IGF-1 is able to increase this levels of fear extinction.

20. The molecular mechanism is not sufficiently addressed. The authors suggest that SK-, m-currents or mGluR5 might be involved. However, no direct evidence for the specific involvement of these channels is provided, nor any explanation for the slow and long-lasting effect observed.

We have analyzed the correlation between the mAHP/sAHP and the percentage of freezing (see new Figure 4F). The correlation parameters are R = 0.702 for mAHP and R = 0.787 sAHP. This correlation is now discussed in the manuscript (page 25): “Moreover, we found that a reduction in AHP amplitude was directly correlated with a reduction in the expression of fear, suggesting that mechanisms favoring a greater AHP reduction would produce an enhancement of the acquisition of extinction memory”.

[Editors’ note: what follows is the authors’ response to the second round of review.]

The manuscript has been improved but there are some remaining issues that need to be addressed, as outlined below:1. The authors performed new experiments showing that inhibition of IGF1Rs occludes the effects of exogenously applied IGF1, but does not cause any effect on neuronal/synaptic function and memory extinction. This raises the question whether endogenous IGF1 plays a physiological role in these processes? In the discussion the authors claim that "....IGF-1 appears as a key endogenous molecule in the modulation of the extinction of conditioned fear memory, supporting the role of IGF-1 as a crucial piece in behavioral tasks." Unfortunately, the results do not support this central claim.We would like to ask the authors to modify these claims and to provide some additional discussion on the relevance of the results to some conditions with elevated cerebral IGF1 levels.

In agreement with the reviewer, we have modified the claim as follows: “Therefore, our results show a novel functional consequence of IGF-1 signalling on animal behavior in the mPFC through the modulation of the extinction of conditioned fear memory”. (page 11, line 215)

We have also provided an additional discussion of the relevance of our results to elevation of cerebral IGF-1 levels observed after running. We have included the following paragraph: “Functional relevance of the IGF-1 modulation in the mPFC. IGF-1 is actively transported to the central nervous system from plasma through the choroid plexus (Carro et al., 2000), and it is also locally produced in the brain by neurons and glial cells (Quesada et al., 2007; Suh et al., 2013; Rodriguez-Perez et al., 2016). Interestingly, the uptake of IGF-1 by the brain correlates with frequency-dependent changes in cerebral blood flow in the cortex during information processing (Nishijima et al., 2010). Therefore, the levels of IGF-1 in the IL would depend on both its active transport from the plasma, favoured by the activity involved in information processing, and in the IGF-1 locally produced by neurons and astrocytes of the IL. Thus, high levels of IGF-1 are expected in the IL because its high activity during the extinction of fear conditioned (Milad and Quirk, 2002; Sepulveda-Orengo et al., 2013). Our results demonstrate that a further increase in the IGF-1 levels due to the exogenous application of IGF-1 in the IL favour the extinction of conditioned fear. In natural conditions, these high levels of IGF-1 could be expected after physical exercises because it is known that IGF-1 uptake from the plasma and IGF-1 levels in the brain are increased after running, reaching the higher levels (Carro et al., 2000). Thus, our results could be related with some of the benefits of physical exercise on the brain function by increasing neuronal excitability and favouring synaptic plasticity and learning and memory through the activation of the IGF-1Rs”. (page 17, line 352)

2. The authors performed new experiments showing that inhibition of G protein activation by GDPβs prevents the reduction in mAHP, while had no effect on sAHP (Fig. 2F-G). First, in the Fig. 2G the mAHP seems to be altered as well. The p-value for this comparison is 0.06, and an additional single cell may render GDPβs ineffective at all at preventing the effect of IGF1 on AHP (6 cells per group seems rather low). Second, the authors wrote in their discussion: "In our study, inhibition of G protein-coupled receptors by GDPβs abolished the IGF-1-mediated reduction of mIAHP, suggesting that the effect of IGF-1 on mAHP could be mediated by the activation of mGluR5." This conclusion is far reaching since there are many other possible mechanisms may explain this effect. The authors should tone down this claim since no evidence on mGluR5 involvement is provided.

We agree with the reviewer. We have increased the number of experiments studying the effect of GDPβs (n = 8) and data confirms that GDPβs prevents the IGF-1-mediated reduction in mAHP (p = 0.109), while had no effect on sAHP reduction (p = 0.023) (Supplementary Table 1).

In agreement with the reviewer, we have tone down this claim by replacing it by the following conclusion: “In our study, inhibition of G protein-coupled receptors by GDPβs abolished the IGF-1-mediated reduction of mIAHP, suggesting that the effect of IGF-1 on mAHP is mediated by the activation of metabotropic receptors. One candidate are the mGluRs, since it has been demonstrated that the activation of mGluR5 increases IL-L5PN excitability (Fontanez-Nuin et al., 2011) and modulates the recall of extinction (Fontanez-Nuin et al., 2011; Sepulveda-Orengo et al., 2013) in a similar way to what we describe for IGF-1. However, we can't rule out the involvement of other GPCRs and a further study is required to analyse the type of metabotropic receptor and the signalling pathway mediating the reduction of the mAHP by the IGF-1”. (page 12, line 246)

3. Figure 5, which is supposed to present mEPSC data, is missing.

The reviewer is completely right. We have now included the missed Figure 5.